# The active sites of Cu–ZnO catalysts for water gas shift and CO hydrogenation reactions

Zhenhua Zhang[1,2,7], Xuanye Chen[1,7], Jincan Kang[3,7], Zongyou Yu[1], Jie Tian[4], Zhongmiao Gong[5], Aiping Jia[1,2], Rui You[1], Kun Qian[1], Shun He[3], Botao Teng[2], Yi Cui [5], Ye Wang [3✉], Wenhua Zhang [1✉] & Weixin Huang [1,6✉]

Cu–ZnO–Al$_2$O$_3$ catalysts are used as the industrial catalysts for water gas shift (WGS) and CO hydrogenation to methanol reactions. Herein, via a comprehensive experimental and theoretical calculation study of a series of ZnO/Cu nanocrystals inverse catalysts with well-defined Cu structures, we report that the ZnO–Cu catalysts undergo Cu structure-dependent and reaction-sensitive in situ restructuring during WGS and CO hydrogenation reactions under typical reaction conditions, forming the active sites of Cu$_{Cu(100)}$-hydroxylated ZnO ensemble and Cu$_{Cu(611)}$Zn alloy, respectively. These results provide insights into the active sites of Cu–ZnO catalysts for the WGS and CO hydrogenation reactions and reveal the Cu structural effects, and offer the feasible guideline for optimizing the structures of Cu–ZnO–Al$_2$O$_3$ catalysts.

[1] Hefei National Laboratory for Physical Sciences at the Microscale, Key Laboratory of Surface and Interface Chemistry and Energy Catalysis of Anhui Higher Education Institutes, School of Chemistry and Materials Science, University of Science and Technology of China, Hefei, China. [2] Key Laboratory of the Ministry of Education for Advanced Catalysis Materials, Institute of Physical Chemistry, Zhejiang Normal University, Jinhua, China. [3] State Key Laboratory of Physical Chemistry of Solid Surfaces, Collaborative Innovation Center of Chemistry for Energy Materials, National Engineering Laboratory for Green Chemical Productions of Alcohols, Ethers and Esters, College of Chemistry and Chemical Engineering, Xiamen University, Xiamen, China. [4] Engineering and Materials Science Experiment Center, University of Science and Technology of China, Hefei, China. [5] Vacuum Interconnected Nanotech Workstation, Suzhou Institute of Nano-Tech and Nano-Bionics, Chinese Academy of Sciences, Suzhou, China. [6] Dalian National Laboratory for Clean Energy, Dalian, China. [7] These authors contributed equally: Zhenhua Zhang, Xuanye Chen, Jincan Kang. ✉email: wangye@xmu.edu.cn; whhzhang@ustc.edu.cn; huangwx@ustc.edu.cn

Since the postulation of the "active site" concept[1], identifications of the active site of a catalyst have always been the Holy Grail of heterogeneous catalysis studies[2–6]. The active site of a catalyst varies with the reaction being catalyzed. Cu-ZnO–Al$_2$O$_3$ catalysts, industrially used as the catalysts for the important water gas shift (WGS) reaction (CO + H$_2$O → CO$_2$ + H$_2$)[7] and CO hydrogenation to methanol reaction (CO + 2H$_2$ → CH$_3$OH)[8,9], are a representative example. This has inspired great efforts devoted to identifying the active sites of Cu–ZnO–Al$_2$O$_3$ catalysts in both reactions. However, debates still exist due to the lack of solid experimental evidence. In the Cu–ZnO based catalysts for the WGS reaction, it is argued whether the metallic copper phase with a unique structure dispersed or stabilized by ZnO[10–12] or the Cu–ZnO interface capable of facilely dissociating H$_2$O[13–16] acts as the active structure. This also led to different reaction mechanisms of Cu–ZnO catalyzed WGS reaction put forward by density function theory (DFT) calculations[12–14], which, however, all did not calculate the activation energy for the H$_2$ formation step. In the Cu–ZnO based catalysts for the CO hydrogenation to methanol reaction, the in situ formed CuZn alloy via the reduction of partial ZnO at defective Cu sites has been proposed as the active phase[17–19], but the structure of defective Cu sites has not been identified. An intimate synergy between Cu and ZnO at the Cu-ZnO interface with ZnO as a structural modifier, hydrogen reservoir, or direct promoter for bond activation was highlighted in the CO$_2$ hydrogenation to methanol reaction with the unavoidable presence of CO[20–22]. Using Cu/MgO model catalysts[19,23], it was demonstrated that Cu nanoparticles supported on irreducible oxide were capable of catalyzing CO hydrogenation to methanol while the ZnO promoter not only greatly enhanced the catalytic activity but also changed the reaction mechanism. Moreover, the active structures of Cu in both types of Cu–ZnO–Al$_2$O$_3$ catalysts are not established.

Uniform nanocrystals (NCs)-based catalytic materials with well-defined structures have demonstrated the successful applications in both fundamental catalysis studies under working conditions and efficient catalyst explorations[24–28]. For examples, the use of uniform cubic, octahedral and rhombic dodecahedral Cu$_2$O NCs that selectively expose the {100}, {111} and {110} facets, respectively, enabled the identification of the active sites of Cu-based catalysts in CO oxidation[29,30], propylene oxidation[31], low-temperature WGS reaction[32] and (photo)catalytic CO$_2$ hydrogenation[33]. In this work, via a combined experimental and theoretical study of various ZnO/Cu–NCs inverse catalysts, we successfully identify the active sites of Cu-ZnO catalysts for WGS and CO hydrogenation reactions respectively as the Cu$_{Cu(100)}$-hydroxylated ZnO ensemble and Cu$_{Cu(611)}$Zn alloy and elucidate the reaction mechanisms, which nicely exemplify the concept of reaction-dependent restructuring and active site of a catalyst.

## Results and discussion

**Synthesis and structural characterizations catalysts.** A series of ZnO/Cu–NCs catalysts were prepared from the corresponding ZnO/Cu$_2$O–NCs catalysts via a morphology-preserved reduction strategy. Uniform capping ligands-free Cu$_2$O NCs, including cubic Cu$_2$O NCs (c-Cu$_2$O) enclosed with {100} crystal planes with different size distributions of 682 ± 92 (denoted as c-Cu$_2$O-682), 109 ± 10 (denoted as c-Cu$_2$O-109), and 34 ± 4.5 (denoted as c-Cu$_2$O-34) nm, octahedral Cu$_2$O NCs enclosed with {111} crystal planes with size distribution of 583 ± 74 nm (denoted as o-Cu$_2$O), and rhombic dodecahedral Cu$_2$O NCs enclosed with {110} crystal planes with size distribution of 550 ± 93 nm (denoted as d-Cu$_2$O) (Fig. 1a1–e1, Supplementary Figs. 1 and 2), were prepared according to well established procedures[29–32] and then used as

the supports to synthesize a series of ZnO/Cu$_2$O-NCs catalysts with preserved morphologies of corresponding Cu$_2$O NCs supports (Fig. 1a2–e2, Supplementary Table 1 and Supplementary Figs. 3–7). The Cu$_2$O NCs and ZnO/Cu$_2$O-NCs catalysts were reduced in 5% CO/Ar at appropriate temperatures chose from CO-temperature-programmed reduction (TPR) results (Supplementary Fig. 8) to acquire corresponding Cu NCs and ZnO/Cu-NCs catalysts that preserve the original morphologies (Fig. 1a3–e3 and a4-e4, Supplementary Figs. 4–7). Only the metallic Cu phase was observed on Cu NCs and ZnO/Cu-NCs catalysts (Supplementary Fig. 9), and electron diffraction patterns indicate that all Cu NCs are single crystals, but the presence of surface Cu(I) species was identified by x-ray photoelectron spectroscopy (XPS) (Supplementary Fig. 10).

As measured by the in-situ diffuse reflectance infrared Fourier transformed spectroscopy (DRIFTS) (Supplementary Fig. 11), ZnO barely adsorbs CO, while Cu$_2$O NCs exhibit two vibrational bands at 2108 and 2120-2145 cm$^{-1}$ assigned to CO adsorbed respectively at the terrace and defective Cu(I) sites[30,34]. c-Cu NCs show two vibrational bands at 2085 and 2101–2106 cm$^{-1}$ arising from CO adsorbed respectively at the terrace and defective sites of Cu{100} facets (Fig. 2a, Supplementary Fig. 12 and Supplementary Table 2)[29,32]. Among all c-Cu NCs, the finest c-Cu-34 NCs exhibit the highest density of defective sites, while c-Cu-109 NCs finer than c-Cu-682 NCs exhibit a lower density of defective sites, which can be associated with different synthesis methods of various c-Cu$_2$O NCs and a lower reduction temperature adopted for the reduction of c-Cu$_2$O-109 NCs. o-Cu NCs show two vibrational bands at 2075 and 2107 cm$^{-1}$ arising from CO adsorbed respectively at the terrace and defective sites of Cu{111} facets (Fig. 2c, Supplementary Fig. 13)[32,35]. d-Cu NCs show one vibrational band at 2093 cm$^{-1}$ arising from CO adsorbed at the terrace sites of Cu{110} facets (Fig. 2d, Supplementary Fig. 13)[35]. The absence of vibrational features for CO adsorbed at the Cu(I) site suggests that the surface Cu(I) species on Cu NCs is O-terminated Cu suboxide (Cu$_x$O, x ≥ 10)[32]. In addition to the vibrational features of CO adsorbed on the Cu surface, all ZnO/Cu-NCs catalysts exhibit a weak vibrational feature at 2130–2137 cm$^{-1}$, characteristic for CO adsorbed at the Cu(I) site (Fig. 1b–d, Supplementary Figs. 12 and 13, Supplementary Table 2). Its intensity initially increases but then decreases as the ZnO loading increases, indicating that the Cu(I) site is located at the ZnO–Cu interface. These assignments are supported by DFT calculation results of vibrational frequencies of CO adsorbed on various Cu surfaces and ZnO-Cu interfaces (Supplementary Fig. 14). The step Cu(211) and Cu(611) surfaces were used to model the common step defects on Cu(111) and Cu(100) surfaces, respectively[17,36]. The calculated vibrational frequency of CO adsorbed on Cu(100) is larger than on Cu(111), and those on the step sites are larger than on the corresponding terrace sites by less than 10 cm$^{-1}$. Resulting from the charge transfer, the copper atom of Cu–O–Zn interface is Cu(I) at which adsorbed CO exhibits a vibrational frequency higher than CO adsorbed on the terrace Cu site by more than 35 cm$^{-1}$. Vibrational features of CO adsorbed at the edge or corner sites were hardly observed, particularly for the c-Cu-34 and ZnO/c-Cu-34 catalysts, indicating that their density should be much lower than the density of face sites.

**Catalytic performance in WGS reaction.** Catalytic performance of various Cu NCs and ZnO/Cu-NCs catalysts were evaluated in the WGS and CO hydrogenation reactions. In the WGS reaction, c-Cu NCs are more active than d-Cu and o-Cu NCs (Supplementary Fig. 15), agreeing with the previous report[32]. Their catalytic activity increases as the sizes decrease whereas the stability decreases. ZnO/Cu NCs show much enhanced catalytic activity and stability than corresponding Cu NCs (Fig. 3a and

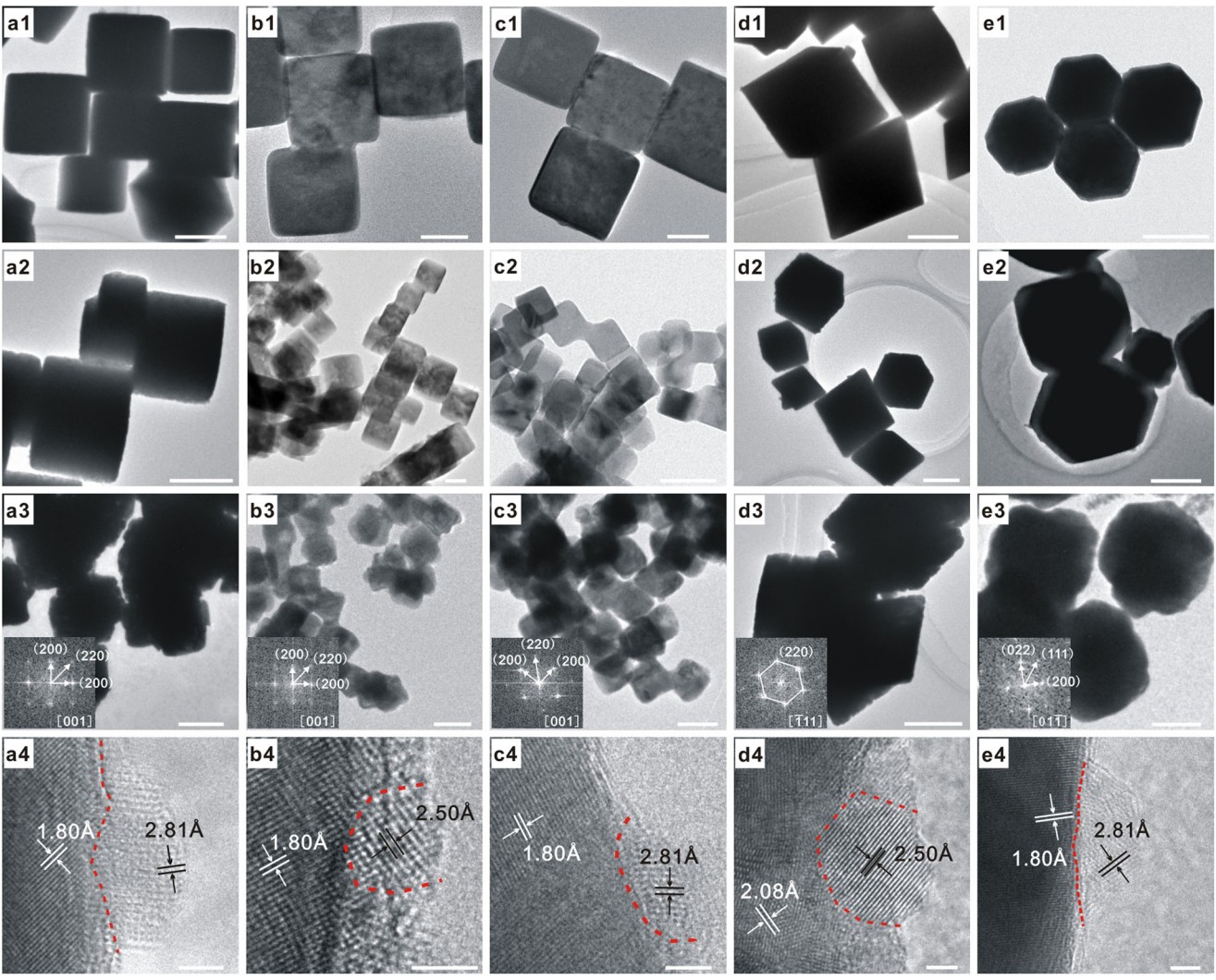

**Fig. 1 Microscopic characterizations.** The scale bars of (**a1**–**a3**), (**d1**–**d3**), and (**e1**–**e3**) correspond to 500 nm, that of **b3** corresponds to 100 nm, those of **b1**, **b2**, **c2**, and **c3** correspond to 50 nm, that of **c1** corresponds to 20 nm, and those of (**a4**–**e4**) correspond to 2 nm. TEM images of as-synthesized **a1** c-Cu₂O-682, **b1** c-Cu₂O-109, **c1** c-Cu₂O-34, **d1** o-Cu₂O, and **e1** d-Cu₂O NCs. TEM images of as-synthesized **a2** 1%ZnO/c-Cu₂O-682, **b2** 1%ZnO/c-Cu₂O-109, **c2** 1%ZnO/c-Cu₂O-34, **d2** 1%ZnO/o-Cu₂O, and **e2** 1%ZnO/d-Cu₂O catalysts. TEM and HRTEM images of as-synthesized (**a3**, **a4**) 1%ZnO/c-Cu-682, (**b3**, **b4**) 1%ZnO/c-Cu-109, (**c3**, **c4**) 1%ZnO/c-Cu-34, (**d3**, **d4**) 1%ZnO/o-Cu, and (**e3**, **e4**) 1%ZnO/d-Cu catalysts. Lattice fringes of 1.80, 2.08, 2.50, and 2.81 Å respectively correspond to the spacing of Cu{200}, Cu{111} (JCPDS card NO. 89-2838), hexagonal ZnO{101}, and ZnO{100} (JCPDS card NO 89-1397) crystal planes. Insets show corresponding electron diffraction patterns of TEM images.

Supplementary Figs. 16–19), and 5%ZnO/c-Cu-109 and 9%ZnO/c-Cu-34 are even more active than the commercial Cu/ZnO/Al₂O₃ WGS catalyst below 423 K, despite of the large Cu particles. Calculated from the corresponding Arrhenius plots (Supplementary Figs. 16-18), all ZnO/c-Cu catalysts show similar apparent activation energies (Eₐ) of 37.7 ± 0.3 kJ/mol and thus exhibit the same type of active site, also indicating that the face sites of Cu NCs in ZnO/c-Cu catalysts dominantly contribute to the catalytic activity, while ZnO/d-Cu, ZnO/o-Cu, Cu/ZnO/Al₂O₃, c-Cu and d-Cu exhibit Eₐ of 40.7 ± 2.6, 55.9 ± 3.9, 51.6 ± 3.7, 54.1 ± 3.1 and 68.4 ± 8.0 kJ/mol, respectively (Fig. 3b). Thus, the Cu{100} facets exposed on c-Cu NCs are the most active facet not only for the Cu catalysts[32] but also for the ZnO/Cu catalysts in the WGS reaction. These results suggest that the high apparent catalytic activity of commercial Cu/ZnO/Al₂O₃ WGS catalyst should result from the density of the active site rather than from the intrinsic activity of the active site. The CO conversions of various ZnO/c-Cu catalysts at 423 K, at which temperature the c-Cu catalysts do not exhibit observable catalytic activity, were found proportional to the amount of CO adsorbed at the Cu(I) site of Cu-O-Zn interface but not to the amounts of other types of adsorbed CO species (Fig. 4a–c and Supplementary Fig. 20). This demonstrates that the low-temperature WGS reaction proceeds at the Cu-ZnO interface of ZnO/c-Cu catalysts.

**Reaction mechanism of WGS reaction.** The temperature-programmed surface reaction (TPSR) spectra of CO and H₂O (Fig. 4d and Supplementary Fig. 21) demonstrate the simultaneous productions of CO₂ and H₂ over ZnO/Cu–NCs catalysts, which occurs at much lower temperatures over ZnO/c-Cu than over ZnO/o-Cu. This further supports that the ZnO/c-Cu catalysts are more intrinsically active than the ZnO/o-Cu catalysts, and meanwhile, demonstrates that the CO₂ and H₂ productions over ZnO/Cu–NCs catalysts result from the same elementary surface reaction or that neither CO₂ production nor H₂ production is the rate-limiting step. However, in the TPSR spectra of CO and H₂O in our previous results over Cu NCs[32], the H₂ production occurs at a higher temperature than the CO₂ production, indicating the H₂ production as

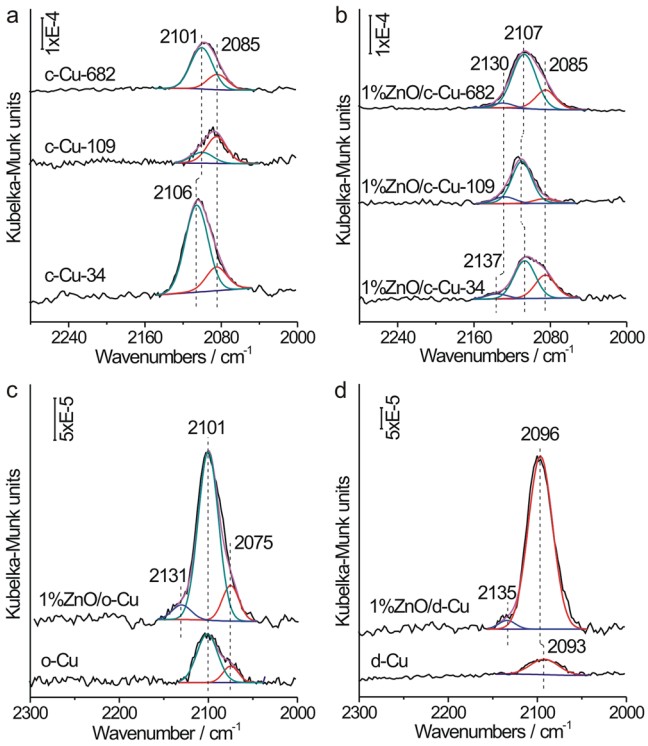

**Fig. 2 Spectroscopic characterizations.** In situ DRIFTS spectra of CO adsorption (($P_{CO}$ = 400 Pa)) at 123 K on **a** various c-Cu NCs, **b** various 1% ZnO/c-Cu catalysts, **c** o-Cu NCs and 1%ZnO/o-Cu catalyst, and **d** d-Cu NCs and 1%ZnO/d-Cu catalyst. The red, green and blue lines represent the fitted vibrational peaks of CO adsorbed on the Cu terrace sites of various facets, defective Cu sites and Cu(I) sites of Cu(I)-O-Zn interface, respectively.

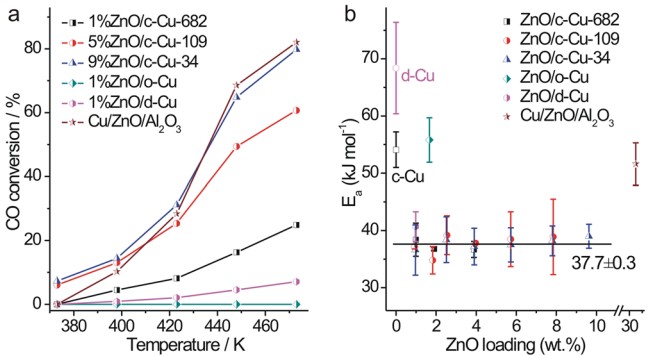

**Fig. 3 Catalytic performance in WGS reaction. a** Catalytic performance of representative ZnO/Cu and commercial Cu/ZnO/Al$_2$O$_3$ WGS catalysts for the WGS reaction; **b** Apparent activation energies ($E_a$) of various catalysts as a function of ZnO loadings.

the rate-limiting step in the Cu-catalyzed WGS reaction with the Cu-Cu$_x$O interface as the active site.

Near-ambient pressure X-ray photoelectron spectroscopy (NAP-XPS) results (Fig. 4e, f, Supplementary Figs. 22 and 23, Supplementary Table 3) show that the Cu and Zn speciation of ZnO/Cu-NCs catalysts do not vary with the reaction condition, suggesting that the WGS reaction catalyzed by the Cu-ZnO interface should not follow the redox mechanism[7]. Hydroxyl and various surface oxygenates, including carboxylate, formate and carbonate, were observed as the surface intermediates. Acquired by CO reduction, 1%ZnO/o-Cu shows a larger OH coverage than

9%ZnO/c-Cu-34 (Fig. 4f), demonstrating the lower reactivity of OH species on 1%ZnO/o-Cu toward CO. The OH species on 9% ZnO/c-Cu-34 during the WGS reaction at 323 K increases at the expense of the O$_{ZnO+CuxO}$ species, corresponding to H$_2$O dissociation at the c-Cu-ZnO interface to produce OH species. With the reaction temperature increasing, the OH coverage keeps decreasing while the oxygenates coverage does not vary much, suggesting that the OH species should be involved in the rate-limiting step but the oxygenates should not. The oxygenates species on 1%ZnO/o-Cu increases at the expense of OH species during the WGS reaction at 323 K, corresponding to the consumption of OH species due to the reaction with gaseous CO to produce surface oxygenates. Their coverage decreases at 423 K and the OH coverage increases, demonstrating the occurrence of surface reactions of oxygenate species and water dissociation; then the coverages of both oxygenates and OH species slightly decrease at 523 K. The coverages of OH and oxygenates surface intermediates are always larger on 1%ZnO/o-Cu catalysts under WGS reaction than on corresponding 9%ZnO/ c-Cu-34 catalysts; meanwhile, the concentrations of gaseous CO and H$_2$O of WGS reaction catalyzed by 1%ZnO/o-Cu are higher than by 9%ZnO/c-Cu-34. These observations demonstrate that the H$_2$O activation are more facile and the formed OH and oxygenates intermediates are more reactive on 9%ZnO/c-Cu-34 than on 1%ZnO/o-Cu, leading to its higher activity in catalyzing the WGS reaction. This is further supported by the NAP-XPS results of 1%ZnO/o-Cu and 9%ZnO/c-Cu-34 catalysts exposed firstly to water and then to CO (Supplementary Figs. 24 and 25, Supplementary Table 4).

**DFT calculations of WGS reaction.** DFT calculations were carried out at the ZnO/Cu(111) and ZnO/Cu(100) surfaces (Fig. 5a1, a2) to explore the WGS reaction mechanism catalyzed by ZnO/ Cu-NCs catalysts (Supplementary Figs. 26 and 27, Supplementary Table 5). Water dissociation into OH$_{Cu}$ and O$_{ZnO}$H at the ZnO–Cu interfaces proceeds very facilely, consistent with previous reports[14,37]. The COOH$_{Cu}$ intermediate formed by adsorbed CO$_{Cu}$ and OH$_{Cu}$ decomposes either to produce gaseous CO$_2$ and O$_{ZnO}$H[38] or to produce gaseous CO$_2$ and H$_{Cu}$[32] with activation energies below 0.65 eV. However, the subsequent H$_2$ production either via the recombinative desorption of two O$_{ZnO}$H groups or via the H transfer from O$_{ZnO}$H group to the Cu site followed by the recombination of two H$_{Cu}$ species need to overcome barriers larger than 1.39 eV. These DFT calculation results will lead to H$_2$ productions at a higher temperature than CO$_2$ productions, against our TPSR experimental results of simultaneous H$_2$ and CO$_2$ productions. Indicated by the DFT calculation results, an accumulation of OH groups can be expected at the Cu-ZnO interface. We thus calculated the activation energy of water dissociation and H transfer reaction at the OH-covered Cu-ZnO interfaces, in which the OH coverage is defined as the ratio of OH number against total O number at the ZnO-Cu interface. The calculated activation energy of water dissociation was found to increase with the OH coverage at the Cu-ZnO interface while the calculated activation energy of H transfer reaction to decrease (Fig. 5c, Supplementary Fig. 28 and Supplementary Table 6), but water dissociation still exhibits smaller activation energy than H transfer reaction at the ZnO-Cu interfaces with OH coverages up to 0.5 ML. When the OH coverage increases to 0.75 ML, the activation energy of H$_2$O dissociation increases to 1.05 and 0.87 eV respectively at the 0.75 ML OH$_{ZnO}$-ZnO-Cu(111) (Fig. 5b1) and 0.75 ML OH$_{ZnO}$-ZnO-Cu (100) (Fig. 5b2) interfaces, larger than the corresponding activation energy of subsequent H transfer reaction, being 0.88 and 0.76 eV, respectively. These DFT calculation results suggest that

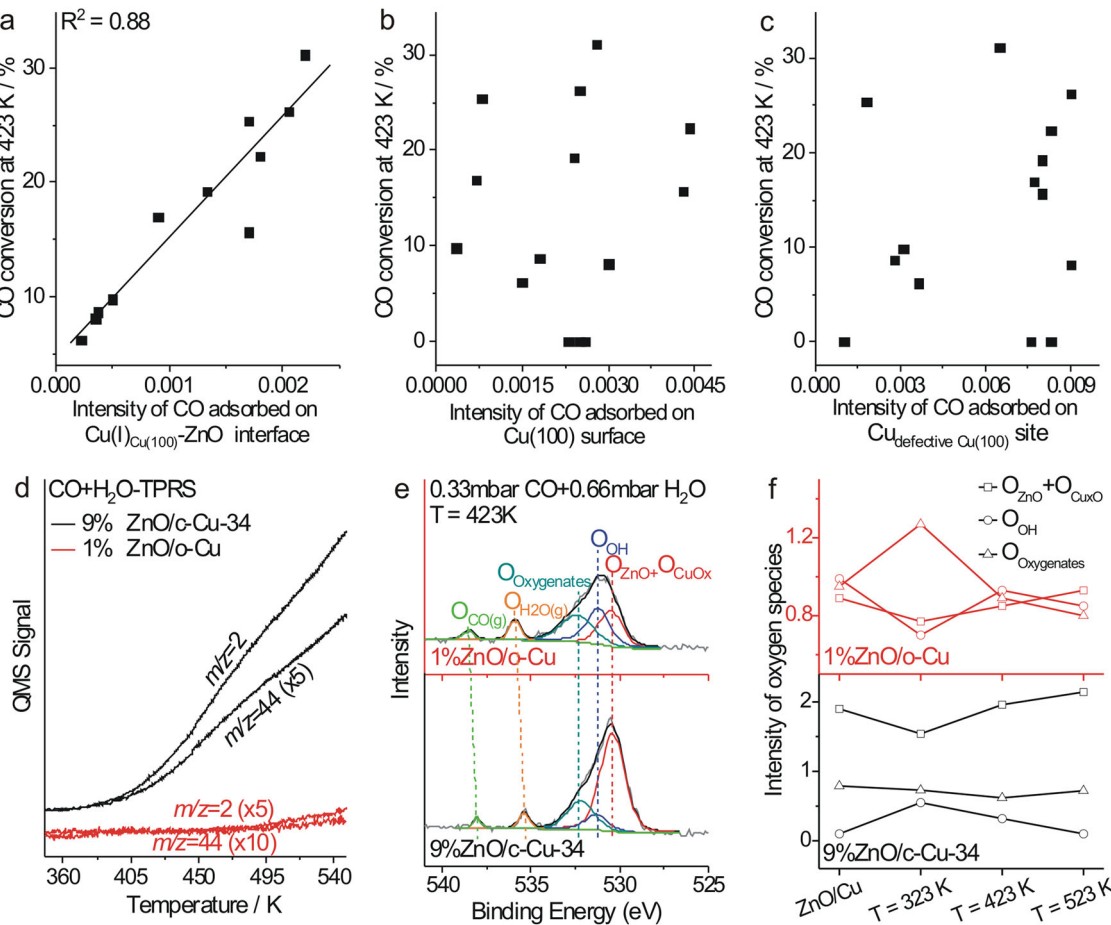

**Fig. 4 Reaction mechanism of WGS reaction.** CO conversion at 423 K as a function of the intensity of CO adsorbed on **a** Cu(I)$_{Cu(100)}$-ZnO interface, **b** Cu (100) surface, and **c** Cu$_{defective\ Cu(100)}$ site derived from corresponding DRIFTS results; **d** Temperature-programmed reaction spectra of WGS reaction over 9%ZnO/c-Cu-34 and 1%ZnO/o-Cu catalysts; **e** O 1s NAP-XPS spectra with peak-fitting results of 9%ZnO/c-Cu-34 and 1%ZnO/o-Cu catalysts under 0.33 mbar CO + 0.66 mbar H$_2$O at 423 K; **f** Variations of the intensity of oxygen-containing species on 9%ZnO/c-Cu-34 and 1%ZnO/o-Cu catalysts under 0.33 mbar CO + 0.66 mbar H$_2$O at different temperatures derived from corresponding NAP-XPS results.

the rate-limiting step of WGS reaction changes from the H transfer reaction, i.e., the H$_2$ production, at the ZnO-Cu interfaces with OH coverages up to 0.5 ML to water dissociation at the ZnO-Cu interface with an OH coverage of 0.75 ML. Meanwhile, all elementary steps proceed more easily at the 0.75 ML OH$_{ZnO}$–ZnO–Cu(100) interface than at the 0.75 ML OH$_{ZnO}$-ZnO-Cu(111) interface (Fig. 5d, Supplementary Fig. 29 and Supplementary Table 7). It can be seen that the calculation results on 0.75 ML OH$_{ZnO}$-ZnO/Cu surfaces agree well with the above experimental observations. Thus, the Cu-hydroxylated ZnO ensemble, instead of Cu–ZnO ensemble, is the active site of Cu/ZnO catalysts to catalyze the WGS reaction, and the Cu$_{Cu(100)}$-hydroxylated ZnO ensemble is more active than the Cu$_{Cu(111)}$-hydroxylated ZnO ensemble. In addition to Cu/ZnO based catalysts, Cu/ZrO$_2$, Cu/CeO$_2$ and Cu/TiO$_2$ were also reported active in the WGS reaction, in which the Cu-oxide ensemble was proposed as the active site[39–42]. Our results of the Cu-hydroxylated ZnO ensemble, instead of the Cu-ZnO ensemble, as the active site of Cu/ZnO based catalysts suggest that further studies are needed to provide combined experimental and theoretical calculation evidence on the rate-limiting step of the WGS reaction, H$_2$O dissociation or H$_2$ production, in order to unambiguously identify the active sites of other Cu/oxides catalysts for the WGS reaction.

**Catalytic performance in CO hydrogenation reaction.** In the steady-state CO hydrogenation reaction at 523 K (Fig. 6a1–a4 and

Supplementary Table 8), o-Cu and c-Cu NCs dominantly yield CH$_4$, agreeing with previous results of unsupported Cu catalysts for CO hydrogenation reaction[43], while d-Cu NCs are inactive. The CO conversion increases as the size of c-Cu NCs decreases. ZnO/d-Cu catalysts are also inactive and ZnO/o-Cu catalysts exhibit similar selectivity to o-Cu NCs, while ZnO/c-Cu catalysts show volcano-shaped dependent CH$_3$OH selectivity on the ZnO loading. The highest CH$_3$OH selectivity among ZnO/c-Cu-682, ZnO/c-Cu-109 and ZnO/c-Cu-34 are 23.4% for 1%ZnO/c-Cu-682, 19.8% for 5%ZnO/c-Cu-109 and 65.9% for 9%ZnO/c-Cu-34, respectively. The catalytic performance of representative 9%ZnO/c-Cu-34 catalyst as a function of reaction time (Supplementary Fig. 30) suggests an in situ formation of active site, leading to increased CO conversion and CH$_3$OH selectivity.

**Reaction mechanism of CO hydrogenation reaction.** The Cu phase in all used Cu NCs and ZnO/Cu-NCs catalysts is metallic (Supplementary Fig. 31), but the used d-Cu NCs and ZnO/d-Cu catalysts were observed to be fully covered with amorphous carbon thin film (Fig. 6b1 and Supplementary Fig. 32), demonstrating the occurrence of serious coking that results in the catalytic inactivity. In addition to originally-existing metallic Cu and ZnO components, CuZn alloy with a size distribution of 3.8 ± 1.0 nm was identified on used ZnO/o-Cu and ZnO/c-Cu catalysts in high-resolution transmission electron microscope (HRTEM) images (Fig. 6b2, b3, Supplementary Figs. 33–37). In the

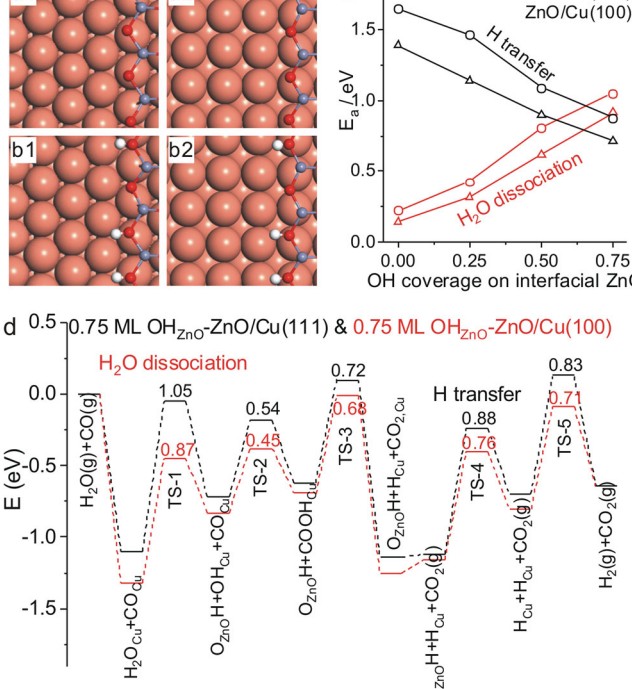

**Fig. 5 DFT calculations of WGS reaction.** Optimized surface structures of **a1** ZnO/Cu(111), **a2** ZnO/Cu(100), **b1** 0.75 ML-OH$_{ZnO}$-ZnO/Cu(111) and **b2** 0.75 ML OH$_{ZnO}$-ZnO/Cu(100). The reddish-orange, purple, red, and white spheres represent Cu, Zn, O, and H atoms, respectively. **c** Calculated activation energy of H$_2$O dissociation and OH$_{ZnO}$-to-H$_{Cu}$ H transfer reaction at ZnO-Cu interface as a function of the OH coverage on interfacial ZnO of ZnO/Cu(111) and ZnO/Cu(100) surfaces. The OH coverage is defined as the ratio of OH number against total O number at the ZnO-Cu interface. **d** Calculated energy profiles of WGS reaction catalyzed by 0.75 ML OH$_{ZnO}$-ZnO/Cu(111) and 0.75 ML OH$_{ZnO}$-ZnO/Cu(100) surfaces.

corresponding x-ray diffraction (XRD) patterns (Supplementary Fig. 38), diffraction peaks of Cu$_2$O and CuO were observed, demonstrating facile oxidation of Cu nanoparticles upon exposures to air, whereas no peaks from CuZn alloy could be identified probably due to their fine size. The percentage of CuZn alloy in Zn-contained components of selected used ZnO/Cu-NCs catalysts was acquired by counting more than 100 particles in the HRTEM images (Fig. 6c) and found to correlate well to the CH$_3$OH selectivity for the ZnO/Cu–NCs catalysts with the same type of Cu NCs support, for example, ZnO/c-Cu-34. This demonstrates the in situ formation of active CuZn alloys in ZnO/Cu catalysts to catalyze CO hydrogenation to CH$_3$OH, consistent with previous reports[17].

The CuZn alloy formation depends on the ZnO loading of ZnO/Cu-NCs catalysts. H$_2$-TPR profiles of various ZnO/Cu-NCs catalysts (Supplementary Figs. 39 and 40) show that the appearance of reduction peak for supported large ZnO particles corresponds to the decrease in the percentage of CuZn alloy. This suggests that highly-dispersed ZnO is more facile to alloy with Cu substrates than ZnO aggregates during CO hydrogenation reaction, agreeing with previous results[18]. Reasonably, the dispersion of supported ZnO in ZnO/c-Cu catalysts increases with the surface area of c-Cu substrate (Supplementary Fig. 41). The CuZn alloy formation also depends on the structure of Cu NCs supports. Not all highly-dispersive ZnO species can form the CuZn alloy. With similar ZnO loadings and Cu NCs sizes, the percentage of CuZn alloy in used 1%ZnO/c-Cu–682 catalyst is much higher than in used 1%ZnO/o-Cu catalyst, suggesting more

facile formation of CuZn alloy on c-Cu NCs than on o-Cu NCs. The percentage of CuZn alloy in different series of used ZnO/c-Cu catalysts does not follow an order of the surface areas of c-Cu NCs, but follows the same order of ZnO/c-Cu-34 > ZnO/c-Cu-682 > ZnO/c-Cu-109 to that of the step site density on various c-Cu NCs (Fig. 2a). These observations indicate that the step sites on c-Cu surfaces are the dominant site for the CuZn alloy formation during CO hydrogenation reaction. The CuZn alloy was previously observed to preferentially form at the step sites of Cu particles in industrial Cu/ZnO/Al$_2$O$_3$ catalyst for CO hydrogenation to CH$_3$OH[17].

In the in situ DRIFTS during CO hydrogenation reaction (Fig. 7a, Supplementary Figs. 42–45 and Supplementary Table 9), 9%ZnO/c-Cu-34 with a high CH$_3$OH selectivity exhibits strong vibrational features of adsorbed CH$_3$O$_a$ and CH$_3$OH$_a$, suggesting the CH$_3$O$_a$ hydrogenation as the rate-limiting step of CO hydrogenation to CH$_3$OH[17,44], while c-Cu-34 with a high CH$_4$ selectivity exhibits significantly-weakened vibrational features of adsorbed CH$_3$O$_a$ and CH$_3$OH$_a$ but vibrational features of adsorbed CH$_2$OH$_a$, CH$_{2,a}$, CH$_{3,a}$ and gaseous CH$_4$, supporting that CO hydrogenation to CH$_4$ proceeds via the CH$_2$OH$_a$ intermediate reacting with adsorbed H$_a$ to form CH$_{2,a}$[45]. An in situ reactor for transmission Fourier transformed-infrared (FT-IR) measurements under conditions varying from high pressures and high temperatures to vacuum and low temperatures was used to firstly characterize various catalysts under CO hydrogenation reaction and then probe their structures by CO adsorption at low temperature without exposures of the used catalysts to air. The acquired in situ transmission FT-IR spectra under CO hydrogenation reaction are similar to the corresponding in situ DRIFTS spectra (Supplementary Fig. 46). During the subsequent CO adsorption (Fig. 7b, Supplementary Fig. 47 and Supplementary Table 10), a vibrational feature at 2075 cm$^{-1}$ arising from Cu{111} facets appears on all used Cu NCs and ZnO/Cu–NC catalysts except on used d-Cu NCs which exhibits no feature due to the entire capsulation by carbon thin film; meanwhile, an additional vibrational band appears at ~2060 cm$^{-1}$ for used ZnO/Cu–NC, which, based on previous[46] and our DFT calculations of CO adsorbed on Zn–Cu(611) and Zn–Cu(211) alloy surfaces (Supplementary Fig. 48), can be assigned to CO adsorbed on CuZn alloy. These observations suggest that only the Cu{111} facets on bare Cu surfaces of used catalysts act to catalyze CO hydrogenation to CH$_4$ under the employed condition while other originally-existing Cu facets and defective sites with low-coordinated Cu atoms on bare Cu surfaces are poisoned by coke formation. The CH$_3$OH formation rates over various ZnO/c-Cu catalysts were found proportional to the amount of CO adsorbed on CuZn alloy of corresponding used catalysts (Fig. 7c, d); meanwhile, the CH$_3$OH formation rate is much smaller over 1%ZnO/o-Cu catalyst than over ZnO/c-Cu catalysts with similar amounts of CO adsorbed on CuZn alloy. This directly demonstrates that CuZn alloy is the active component of ZnO/Cu-NC catalysts to catalyze CO hydrogenation to CH$_3$OH and more active CuZn alloy form on ZnO/c-Cu catalysts than on ZnO/o-Cu catalysts.

CO$_2$ was always produced via the WGS reaction at the Cu-hydroxylated ZnO sites during CO hydrogenation reaction over our ZnO/Cu-NCs catalysts, in which H$_2$O resulted from the reactions of CO hydrogenation to hydrocarbons. CO$_2$ hydrogenation was reported to proceed faster than CO hydrogenation to produce methanol over Cu/ZnO/Al$_2$O$_3$ catalysts[19]. However, CO hydrogenation is the dominant pathway to produce methanol over our ZnO/Cu-NCs catalysts. On one hand, the amount of produced CO$_2$ was significantly less than that of CO in the reaction atmosphere; on the other hand, the in situ DRIFTS spectra (Fig. 7a) only observed the CH$_3$O$_a$ species but not formate species, the key intermediate respectively in CO and CO$_2$ hydrogenation pathways[17,19].

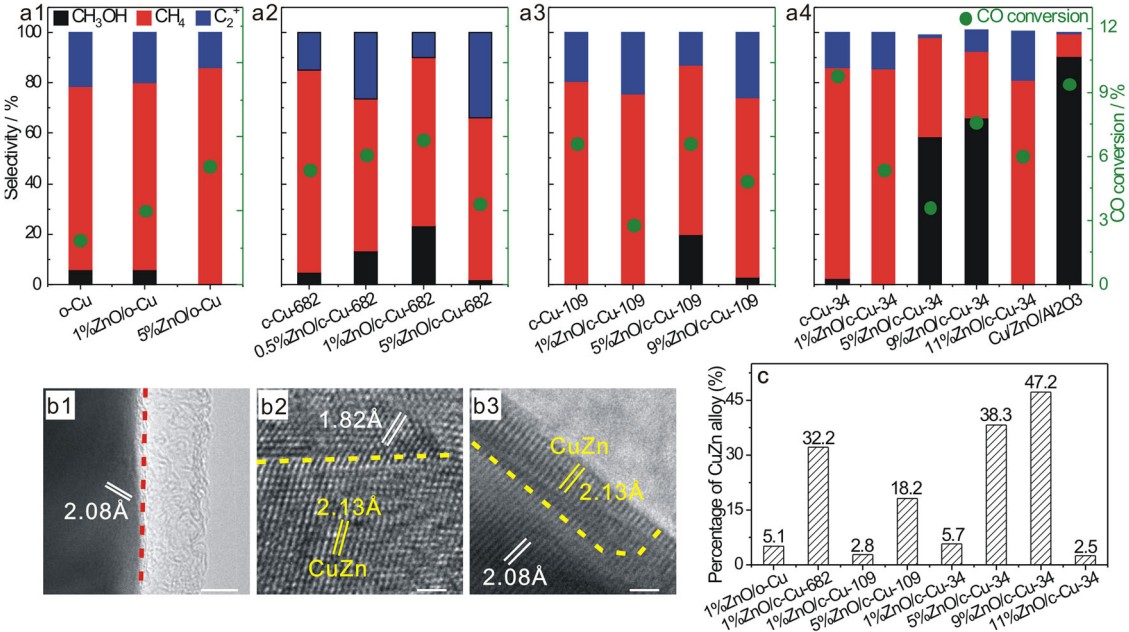

**Fig. 6 Catalytic performance in CO hydrogenation reaction and microscopic characterizations.** The scale bar of **b1** corresponds to 5 nm and those of **b2** and **b3** correspond to 1 nm. **a1–a4** Catalytic performance of various Cu and ZnO/Cu catalysts in the CO hydrogenation to methanol reaction. $CO_2$ is not included for selectivity calculations. Representative HRTEM images of used **b1** 1%ZnO/d-Cu, **b2** 1%ZnO/c-Cu-682, and **b3** 1%ZnO/o-Cu catalysts; **c** Statistical percentage of CuZn alloy nanoparticles of various ZnO/Cu catalysts derived from corresponding HRTEM images. Lattice fringes of 1.82, 2.08, and 2.13 Å respectively correspond to the spacing of Cu{100}, Cu{111} (JCPDS card NO. 89-2838), and ZnCu alloy {111} (JCPDS card NO 89-1397) crystal planes.

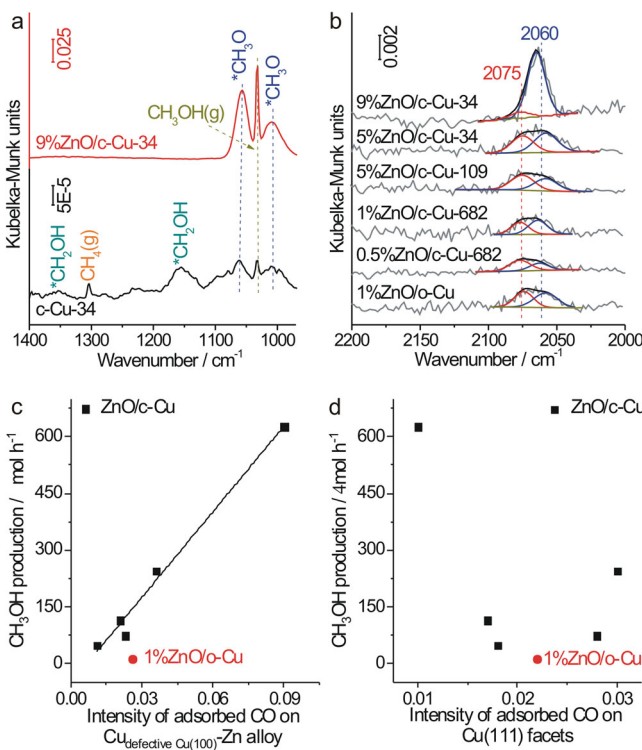

**Fig. 7 Reaction mechanism of CO hydrogenation reaction. a** In situ DRIFTS spectra of c-Cu-34 and 9%ZnO/c-Cu-34 catalysts under 33.3% CO + 66.7% $H_2$ (total pressure: 2 MPa) at 523 K. **b** In situ transmission FT-IR spectra of CO adsorption ($P_{CO}$ = 1000 Pa) at 123 K on various used ZnO/Cu catalysts in CO hydrogenation to methanol reaction without exposure to air, and the red and blue lines represent the fitted vibrational peaks of CO adsorbed on the Cu{111} facets and CuZn alloy, respectively. $CH_3OH$ productions over various ZnO/Cu catalysts as a function of the intensity of CO adsorbed on **c** $Cu_{defective\ Cu(100)}$-Zn alloy and **d** Cu{111} facets derived from corresponding DRIFTS results.

**DFT calculations of CO hydrogenation reaction.** The activation energy for CuZn alloy formation of ZnO on various Cu surfaces was calculated to follow an order of ZnO/Cu(611) < ZnO/Cu (211) < ZnO/Cu(100) < ZnO/Cu(111) (Fig. 8a and Supplementary Fig. 49), agreeing with the experimental results that the step sites on Cu NCs are the dominant site for CuZn alloy formation and the CuZn alloy formation is more facile on the step sites of c-Cu NCs than of o-Cu NCs. Meanwhile, the calculated reaction mechanism of CO hydrogenation to $CH_3OH$ (Fig. 8b, Supplementary Fig. 50 and Supplementary Table 11) demonstrates that adsorbed $CH_3O_a$ hydrogenation is the rate-limiting step and proceeds with a smaller activation energy on Zn-Cu(611) alloy surface than on Zn-Cu(211) alloy surface, consistent with the experimental observations of $CH_3O_a$ and $CH_3OH_a$ as the surface intermediates and of a higher catalytic activity of CuZn alloy formed on ZnO/c-Cu than on ZnO/o-Cu. Thus, $Cu_{Cu(611)}Zn$ alloy is the active site of Cu-ZnO catalysts for catalyzing CO hydrogenation reaction to $CH_3OH$. Compared with industrial Cu/ZnO/$Al_2O_3$ catalysts, the density of surface Cu atoms of our ZnO/Cu-NCs catalysts are much less, leading to the decreased CO conversion; meanwhile, the density of defective Cu sites beneficial for the CuZn alloy formation are much less, leading to the decreased methanol selectivity, because the existing bare Cu sites and Cu-hydrogenated ZnO sites respectively catalyze the CO hydrogenation to hydrocarbons and WGS reactions.

Thus, the ZnO/Cu catalysts undergo different in situ restructuring processes during WGS and CO hydrogenation reactions under typical reaction conditions to form the Cu-hydroxylated ZnO ensemble and CuZn alloy active sites, respectively. These results demonstrate reaction-sensitive restructuring and active sites of Cu-ZnO catalysts. Moreover, the in situ restructuring processes are modulated by the Cu structure to form the active sites with highest intrinsic catalytic activity, $Cu_{Cu(100)}$-hydroxylated ZnO ensemble and $Cu_{Cu(611)}Zn$ alloy for WGS and CO hydrogenation reactions, respectively. Considering the identified active sites, fabricating Cu-ZnO catalysts with Cu{100} facets as many as possible and with Cu

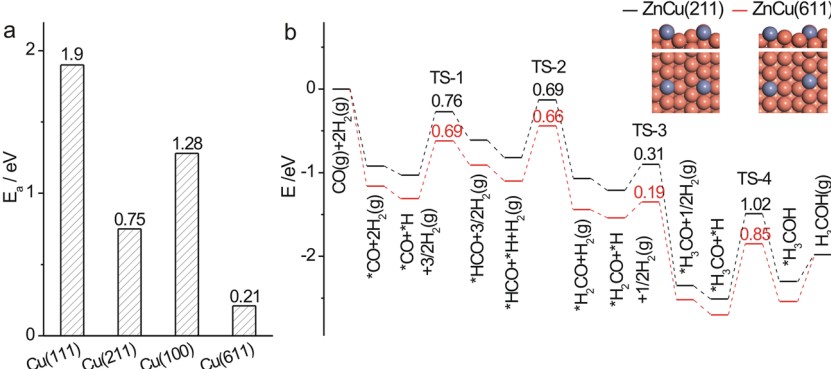

**Fig. 8 DFT calculations of CO hydrogenation reaction. a** Calculated activation energy for CuZn alloy formation of ZnO on various Cu surfaces and **b** calculated energy profiles of CO hydrogenation into methanol by ZnCu(211) and ZnCu(611) alloys. Insets show the optimized surface structures of ZnCu (211) and ZnCu(611) alloys. The reddish-orange and purple spheres represent Cu and Zn atoms, respectively.

{611} step sites as many as possible are effective strategies to develop highly efficient Cu-ZnO catalysts for water gas shift and CO hydrogenation reactions, respectively.

## Methods

**Materials**. All chemicals were purchased from Sinopharm Chemical Reagent Co., Ltd. and used without further purification. 5% CO/Ar, 5% $H_2$/Ar, 66.7% $H_2$/33.3% CO, 0.432% CO/Ar, $O_2$ (99.999%), Ar (99.999%), $C_3H_6$ (99.95%), $N_2$ (99.999%), and CO (99.99%) were purchased from Nanjing Shangyuan Industrial Factory and used without further purification. Ultrapure water (>18.5 MΩ) was used.

**Catalyst synthesis**. $Cu_2O$ cubes with size distribution of 682 ± 92 nm and octahedra with size distribution of 583 ± 74 nm were synthesized according to Zhang et al.'s method[47]. Typically, 5 mL of NaOH aqueous solution (2.0 mol L$^{-1}$) was added dropwise into 50 mL of $CuCl_2$ aqueous solution (0.01 mol L$^{-1}$) at 328 K under continuous stirring ($Cu_2O$ octahedra containing 4.44 g of poly(vinylpyrrolidone) (PVP)). After stirring for 0.5 h, 5 mL of ascorbic acid (AC) solution (0.6 mol L$^{-1}$) was subsequently added dropwise into the solution. The mixture was stirred at 328 K for additional 5 h. The acquired precipitate was collected by centrifugation, decanted by repeated washing with ultrapure water and absolute ethanol for several times, and finally dried in vacuum at room temperature for 12 h. The acquired cubic and octahedral $Cu_2O$ nanocrystals (NCs) were denoted as c-$Cu_2O$-682 and o-$Cu_2O$–PVP, respectively.

$Cu_2O$ cubes with size distributions of 109 ± 10 and 34 ± 4.5 nm were synthesized according to Chang et al.'s method[48]. Typically, 1 mL of $CuSO_4$ aqueous solution (1.2 mol L$^{-1}$) was rapidly added into 400 mL ultrapure water at 298 K ($Cu_2O$ cubes with size distribution of 109 ± 10 nm containing 0.26 g of sodium citrate). After stirring for 5 min, 1 mL of NaOH aqueous solution (4.8 mol L$^{-1}$) was added into the solution. The solution color turned from clear blue solution immediately to turbid blue, indicating $Cu(OH)_2$ formed. After stirring for another 5 min, 1 mL of AC aqueous solution (1.2 mol L$^{-1}$) was added as a reducer and the resulting solution was maintained for additional 0.5 h at 298 K. The solution color gradually turned from turbid blue to yellowish brown. The acquired precipitate was collected by centrifugation, decanted by repeated washing with ultrapure water and absolute ethanol for several times, and finally dried in vacuum at room temperature for 12 h. The acquired $Cu_2O$ NCs with size distributions of 109 ± 10 and 34 ± 4.5 nm were denoted as c-$Cu_2O$-109 and c-$Cu_2O$-34, respectively.

$Cu_2O$ rhombic dodecahedra with size distribution of 550 ± 93 nm were synthesized according to Liang et al.'s method[49]. Typically, 4 mL of oleic acid (OA) mixed with 20 mL of absolute ethanol was added into 40 mL of $CuSO_4$ aqueous solution (0.025 mol L$^{-1}$) under continuous stirring at 373 K. Then, 10 mL of NaOH aqueous solution (0.8 mol L$^{-1}$) was added. After stirring for 5 min, 30 mL of D-( + )-glucose aqueous solution (0.63 mol L$^{-1}$) was added. The resulting mixture were further stirred at 373 K for 1 h to acquire a brick-red precipitate. Centrifugation, decantation by repeating washing with ultrapure water and absolute ethanol for several times were performed, and the final precipitate was dried in vacuum at room temperature for 12 h. The acquired rhombic dodecahedral $Cu_2O$ NCs were denoted as d-$Cu_2O$-OA.

The removal of capping ligands (PVP on o-$Cu_2O$-PVP and OA on d-$Cu_2O$-OA) followed a controlled oxidation procedure developed by Hua et al.[31]. Typically, as-synthesized $Cu_2O$ NCs (ca. 0.2 g) were placed in a U-shaped quartz microreactor. The atmosphere was first purged by the stream of $C_3H_6 + O_2 + N_2$ mixture ($C_3H_6$:$O_2$:$N_2$ = 2:1:22, total flow rate: 50 mL min$^{-1}$) at room temperature for 30 min. Then, the sample was heated to the desirable temperature at a rate of 5 K·min$^{-1}$ (o-$Cu_2O$-PVP: 473 K; d-$Cu_2O$-OA: 488 K) and kept for 30 min. Next, the steam was switched to high-pure Ar (flow rate: 30 mL min$^{-1}$) in which the sample was naturally cooled to room temperature. The acquired $Cu_2O$ octahedra

and rhombic dodecahedra without capping ligands were denoted as o-$Cu_2O$ and d-$Cu_2O$, respectively.

ZnO supported on $Cu_2O$ NCs (ZnO/$Cu_2O$) catalysts were synthesized via incipient wetness impregnation method. Typically, 200 mg of as-synthesized $Cu_2O$ NCs were adequately incipient wetness impregnated with calculated amounts of zinc nitrate ($Zn(NO_3)_2$·$6H_2O$) aqueous-ethanol mixture solution. The resulting sample was dried in vacuum at room temperature for 12 h and then heated at 623 K in high-pure Ar with a flow rate of 50 mL·min$^{-1}$ for 2 h to prepare ZnO/$Cu_2O$ catalysts. The acquired ZnO/$Cu_2O$ catalysts were further reduced in 5% CO/Ar with a flow rate of 30 mL min$^{-1}$ at appropriate temperatures chose from CO-TPR results for 2 h to prepare corresponding ZnO/Cu catalysts. The reduction temperatures are 548 K for c-$Cu_2O$-682, ZnO/c-$Cu_2O$-682, o-$Cu_2O$, ZnO/o-$Cu_2O$, d-$Cu_2O$, and ZnO/d-$Cu_2O$, and 473 K for c-$Cu_2O$-109, c-$Cu_2O$-109, ZnO/c-$Cu_2O$-34 and ZnO/c-$Cu_2O$-34.

Details on structural characterizations, activity evaluations, and DFT calculations can be found in the supplementary information.

## Data availability

The data supporting the findings of the study are available within the paper and its supplementary information. Source data are provided with this paper.

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

## Acknowledgements

This work was financially supported by the National Key R & D Program of MOST (2017YFB0602205), the National Natural Science Foundation of China (91945301, 21525313, 91745202), the Chinese Academy of Sciences, and the Changjiang Scholars Program of Ministry of Education of China. W.Z. is supported by USTC Tang Scholarship. The DFT calculations were performed on the supercomputing center of University of Science and Technology of China (USTC-SCC) and of Guangzhou (Guangzhou-SCC). We thank Xianquan Industrial and Trading Co., Ltd. (Tianjin, China) for assistance on in situ transmission FT-IR measurements under conditions varying from high pressures and high temperatures to vacuum and low temperatures.

## Author contributions

W.H. designed and supervised the project. W.Z. supervised the DFT calculations. Y.W. supervised the catalytic activity evaluation in CO hydrogenation. Z.Z. carried the catalyst preparation and characterization, and catalytic activity evaluation in WGS reaction. X.C. carried out the DFT calculations. J.K carried out the catalytic activity evaluation in CO hydrogenation. Z.Y., J.T., Z.G., A.J., R.Y., K.Q., S.H., B.T., and Y.C. assisted with the experiments. All authors analyzed the data. W.H., Z.Z., and X.C. prepared the manuscript and other authors commented on the manuscript.

## Competing interests

The authors declare no competing interests.
