## [Peer Review File · Nature Communications]

REVIEWER COMMENTS

Reviewer #1 (Remarks to the Author):

In the manuscript ZnO/Cu nanocrystal inverse catalysts with different defined Cu structures were synthesized and the influence of the Cu surface planes was investigated for the WGS and the hydrogenation of CO to methanol. The authors found a restructuring of the catalyst surface for both reactions leading to different active surface sites. For the WGS the in situ formed Cu(100)-ZnOH ensemble was found to be the active site, whereas a CuZn alloy formed at the defective Cu(611)Zn interface was found to be the active site for CO hydrogenation. The manuscript only partially offers new insight in catalysis over Cu/ZnO catalysts. While the active site of an in situ formed Cu(100)-ZnOH ensemble is a novel interesting theory, the formation of a CuZn alloy for CO hydrogenation has been previously described in literature.[1] Furthermore, not all important figures were included in the text and were shifted to the SI part (e.g. Figure S20). Moreover, too many graphs were combined in one figure making it hard to read. Therefore, the manuscript may become suitable for publication after major revisions, although I doubt that this is possible.

1) The formation of an Cu(100)-ZnOH ensemble as the active site for the WGS provides interesting new insight in the WGS.

However, this article does not explain why other ZnO-free Cu/metal-oxide systems, such as Cu/ZrO₂, Cu/CeO₂ Cu/TiO₂ also provide a high activity in the WGS[2,3]

2) The authors investigate the CuZnO system for the hydrogenation of CO. However, it is not mentioned that in literature other catalytic systems such as Cu/MgO are described to be more suitable for the CO₂ free synthesis of methanol.[1,4] To study the electronic promotion of ZnO and to separate this from the support effect ZnO introduces in the classical CuZnAl system, a model system with Cu nanoparticles supported on irreducible MgO was developed by Studt et al.[1] The Cu/MgO catalyst showed a high methanol formation rate in a CO/H₂ mixture. This result confirms that Cu is, in general, capable of converting CO to methanol at industrially relevant rates. Furthermore, the addition of 5 wt% ZnO to the Cu/MgO catalyst by impregnation provided evidence that the ZnO promoter plays a decisive role for the catalytic properties of Cu in industrial catalysts, as was observed for model catalysts. Comparing the different catalyst, here DFT calculations already showed that when Zn is introduced into the Cu steps all intermediates and transition states that bind through an oxygen atom are stabilized.

3) In the manuscript the authors describe the formation of an active site for hydrogenation of CO towards methanol at the interface of an defective Cu(100) surface and Zn leading to an alloy. The description of the active site as an alloy of a defective Cu surface and Zn has already been described in literature.[5,6] It is reported to be the active site for the synthesis of methanol from hydrogenation of CO as well as CO₂ over Cu/ZnO/Al₂O₃ catalyst, however CO₂ hydrogenation is described to be significantly faster over Cu/ZnO based catalysts.[1]

4) Since CO₂ is one of the major byproducts produced in the given work (Table S7), the formation of CO₂ and the influence of CO₂ has to be discussed in the manuscript, but it is not mentioned. Also, the origin

of CO₂ is not mentioned at all.

5) Not only the conversion of CO, but also the selectivity towards methanol is significantly lower compared with the industrial methanol synthesis catalyst for all synthesized catalysts. A lower conversion is comprehensible due to the lower amount of Cu surface atoms, however is there a suitable explanation for the difference in the selectivity?

6) Information about the progress of CO conversion over time for the CO hydrogenation similar to the information given for the WGS (Figure S14- Figure S19) would also add additional information about how fast the restructuring happens and about the stability of the catalyst.

7) Furthermore, the pressure of 2 MPa applied for methanol synthesis is significantly lower compared to the pressure of 5-10 MPa, which is usually used in industry leading to a certain lack of comparability to relevant conditions.

Further notes:

- Although the authors claim plausibly that the interface between defective Cu and Zn is most important for the activity in CO hydrogenation, information about the Cu-surface area for the investigated catalysts is crucial. Metallic Cu plays a major role for activation of H₂ via dissociative adsorption. Additionally, by comparing with the BET surface area before the addition of ZnO, the Cu surface area indirectly yields information about the dispersion of the added amount of ZnO.
- The manuscript did not manage to include all relevant figures in the text in a manner it is still possible to read. Since important figures were placed in the SI part (e.g. Fig. S45), while way too much focus was laid on e.g. TEM images (Fig. 1) the manuscript is only barely readable.
- XPS and IR graphs are normally plotted from high to low values.
- Line 132 and 136: Figure 2C and 2D are mixed up.
- Line 214: It is not straightforward to plot two different XPS measurements within the same y axis. For a direct comparison the intensity, which depends on various factors like position of the sample inside the chamber varies too much between two consecutive measurements.
- Line 217: The authors discuss the selectivity of methanol formation, whereas the respective figure (Figure 4F) only provides information about the productivity.
- Line 238: Please discuss that there is a significant pressure gap between the catalytic measurements and the DRIFTS and in situ transmission FT-IR spectra. Additionally, even if the pressure is mentioned in Fig. 4D, in the figures and table in the supportive information a pressure is not noted.
- Line 251: Describing Fig. 4E and Fig. S43 please add a naming of the different fits in blue, red and light blue.
- Line 261: Since the reliability the quantification of IR spectra strongly depends on the chosen region due to possible overlapping bands, here a table of the chosen limits while integrating the bands for Fig. 4F and S45 would be straightforward.
- Line 424: This paper is from 2020 not 2011.
- Figure 4A is misleading, because the selectivities of CO₂ are not included.

- SI Line 370: Fig. S14: “most stable CO species adsorbed on...”

[1] F. Studt, M. Behrens, E. L. Kunkes, N. Thomas, S. Zander, A. Tarasov, J. Schumann, E. Frei, J. B. Varley, F. Abild-Pedersen et al., *ChemCatChem* 2015, 7, 1105–1111.

[2] C. Chen, C. Ruan, Y. Zhan, X. Lin, Q. Zheng, K. Wei, *Int. J. Hydrogen Energy* 2014, 39, 317–324.

[3] J. A. Rodriguez, P. Liu, X. Wang, W. Wen, J. Hanson, J. Hrbek, M. Pérez, J. Evans, *International Symposium on Catalysis for Ultra-Clean Fuels*, Dalian, China, July 21-24, 2008 2009, 143, 45–50.

[4] N. D. Nielsen, J. Thrane, A. D. Jensen, J. M. Christensen, *Catal Lett* 2020, 150, 1427–1433.

[5] F. Studt, M. Behrens, F. Abild-Pedersen, *Catal. Lett. (Catalysis Letters)* 2014, 144, 1973–1977.

[6] M. Behrens, F. Studt, I. Kasatkin, S. Kühl, M. Hävecker, F. Abild-Pedersen, S. Zander, F. Girgsdies, P. Kurr, B.-L. Kniep et al., *Science* 2012, 336, 893–897.

Reviewer #2 (Remarks to the Author):

The current study reports the mechanistic understanding of active sites for WGS and CO hydrogenation using the combined experiment-theory studies. Major revisions are suggested to be made to improve the quality and merit the publication in *Nature Commun.*

First of all, the catalytic process selected here is different from that which has been extensively studied previously for Cu-ZnO: CO₂ hydrogenation to methanol. This is the author-called “long-existing debate” with lots of cited papers. The change from CO₂ to CO hydrogenation imposes different effects on reaction mechanism and surface structures, which are the focus here. Thus, to target the “long-existing debate”, as claimed, CO₂ rather than CO hydrogenation should be studied. Otherwise, the provided understanding may not be meaningful.

For CO adsorption on Cu, no CO-Cu(1+) interaction was observed by IR. Is it possible that CO results in the reduction of Cu(1+) to Cu(0), which may occur on the surface?

In DFT calculations, a nanorow of ZnO was deposited on Cu(111). Was there any strain artificially introduced on ZnO layer to accommodate the lattice mismatch between ZnO and Cu? How will that affect the results? Is the nanorow good enough to describe the experimental samples?

How was *OH coverage defined? In DFT calculations, it seems to be based on the number of Zn sites, which is 0.75ML. I am not sure how 0.75ML of *OH on ZnO was determined. Is it consistent with experiment? Will the coverage of *OH change the catalytic behaviors?

Reviewer #3 (Remarks to the Author):

The present paper by Zhang et al. is an interesting study of the active sites of Cu/ZnO catalysts in the WGS reaction and in the methanol formation by CO hydrogenation. The authors combine an elegant synthetic approach, namely the preparation of Cu₂O pre-catalyst with defined shape and termination that were after the addition of ZnO transformed into Cu metal while maintaining the particle morphology of the inverse catalyst type. Many complementary techniques have been used to explain the catalytic data. The major conclusions of the regarding the active sites are based on CO-DRIFTS and DFT calculation. The results are certainly of high interest to the catalysis community and deserve publication.

However, I do not see that these results “conclude the long-existing debate” on the active sites of this catalyst as claimed by the authors in the abstract. Both the special Cu-ZnO perimeter site and the stepped CuZn alloy site were suggested previously, and this paper presents support for their role in WGS and methanol synthesis, respectively. The suggested Cu-hydroxylated ZnO at Cu(100) ensembles and the CuZn at Cu(661) alloy may be new specific suggestions in the debate, but the experimental evidence presented for their existence compared to other possible sites on the non-uniform powders is rather limited.

Therefore, I recommend revising the final claim of unambiguous identification of these sites and submission to a journal specialized on chemistry or catalysis.

Before re-submission, I recommend to consider the following aspects when revising the paper:

- Some of the reactivity data deviates from expectations for Cu-based catalysts. For example, the dominant formation of methane or the high selectivity to C₂+ products in the methanol synthesis on Cu/ZnO are surprising. This could lead to the conclusion that the catalyst prepared are substantially different from industrial catalysts and may not be suitable models for the elucidation of the active sites of Cu/ZnO/Al₂O₃, which shows almost 100% selectivity to methanol.
- Why were the edges or corners of the particles not considered as possible active sites? Given the established structure-sensitivity of both reactions, these seem likely candidates, which should not be neglected in the discussion of the active sites and in the assignment of CO chemisorption data.
- Activity data for the methanol synthesis should be provided. CO hydrogenation on CuZn is typically much slower than CO₂ hydrogenation that could happen in CO/CO₂/H₂ syngas, which should be discussed.
- Please provide experimental evidence that not only the particle morphology but also the single crystalline nature of the particle was maintained when reducing Cu₂O, e.g. by electron diffraction like shown in Fig. S1 for the Cu₂O. For example, in Figure 1 A4, B4, the 200 lattice planes seem not oriented in parallel or perpendicular to the cube's 100 surface.
- Even if 100 particles were evaluated in HRTEM, the alloy formation and its extent should be confirmed

and quantified by XRD, which should be well suited for this purpose.

- The Miller index of the first reflection in Fig. S2 is wrong.

Authors' Reply to Reviewer 1's Comments and Revisions

Comment: In the manuscript ZnO/Cu nanocrystal inverse catalysts with different defined Cu structures were synthesized and the influence of the Cu surface planes was investigated for the WGS and the hydrogenation of CO to methanol. The authors found a restructuring of the catalyst surface for both reactions leading to different active surface sites. For the WGS the in situ formed Cu(100)-ZnOH ensemble was found to be the active site, whereas a CuZn alloy formed at the defective Cu(611)Zn interface was found to be the active site for CO hydrogenation. The manuscript only partially offers new insight in catalysis over Cu/ZnO catalysts. While the active site of an in situ formed Cu(100)-ZnOH ensemble is a novel interesting theory, the formation of a CuZn alloy for CO hydrogenation has been previously described in literature.[1] Furthermore, not all important figures were included in the text and were shifted to the SI part (e.g. Figure S20). Moreover, too many graphs were combined in one figure making it hard to read. Therefore, the manuscript may become suitable for publication after major revisions, although I doubt that this is possible.

Author Reply: We appreciate the reviewer's positive recommendation and valuable comments very much. We are grateful to the reviewer's comments that the active site of an in situ formed Cu(100)-ZnOH ensemble for WGS reaction is a novel interesting theory. Meanwhile, we also agree with the reviewer's comments that the in situ formation of active CuZn alloy via defective Cu interacting with the reduction of partial ZnO has been proposed for CO hydrogenation in literature, as we described in the introduction part of our original manuscript with cited references (Ref. 16 and 17: Science 336, 893-897 (2012) and Science 352, 969-974 (2016)). With this regard, the novelty of our work is that, by using various Cu nanocrystals with well-defined structures as the supports, we successfully identified that CuZn alloy is formed more facile on the defective sites of Cu(100) surface than of Cu(111) surface and meanwhile is more catalytic active in CO hydrogenation. Therefore, our work not only identify the structures of in-situ formed active sites of Cu-ZnO interface in the WGS and CO hydrogenation reactions, but also demonstrate the effect of Cu structure in the in-situ formed active sites. We believe that these results are of importance and broad interest in both fundamental catalysis and industrial catalysis because Cu-ZnO based catalysts are used as industrial catalysts for the WGS and CO hydrogenation reactions.

In reply to the reviewer, we have clarified this issue in the revised manuscript as following: **"In the Cu-ZnO based catalysts for the CO hydrogenation to methanol reaction, the in situ formed CuZn alloy via the reduction of partial ZnO at defective Cu sites has been proposed as the active phase, ¹⁷⁻¹⁹ but the structure of defective Cu sites has not been identified."** (Please see Lines 50-53). Meanwhile, we added the paper recommended by the reviewer (*ChemCatChem* 7, 1105-1111 (2015)) as Ref. 19 and re-ordered the references accordingly.

We have also accepted the reviewer's suggestion to re-plot all figures in the revised manuscript so that they include all important information and are easy to read.

We have also seriously considered the reviewer's other comments and revised our manuscript accordingly. We hope that the revised manuscript will be suitable for the publication in *Nature Communications*.

Comment 1. The formation of an Cu(100)-ZnOH ensemble as the active site for the WGS provides

interesting new insight in the WGS. However, this article does not explain why other ZnO-free Cu/metal-oxide systems, such as Cu/ZrO₂, Cu/CeO₂, Cu/TiO₂ also provide a high activity in the WGS [2,3].

Author Reply: We appreciate the reviewer's insightful comments very much. In our case, we experimentally observed the simultaneous production of CO₂ and H₂ in the CO+H₂O-TPRS profiles over ZnO/Cu catalysts (Fig. 4d in the revised manuscript), which demonstrates that the CO₂ and H₂ productions over ZnO/Cu-NCs catalysts result from the same elementary surface reaction or that neither CO₂ production nor H₂ production is the rate-limiting step. However, our DFT calculation results using the ZnO-Cu interface showed that H₂ production is the rate-limiting step, inconsistent with the experimental observations. This led us to establish the models of hydroxylated ZnO-Cu interface and eventually identify 0.75 ML OH_{ZnO}-ZnO-Cu interface as the active site of Cu-ZnO catalysts for the WGS reaction.

Interestingly, we have looked into the references on Cu-oxides catalysts for WGS very carefully, but found that few experimental work did the TPRS experiments while all DFT calculations always focused on the dissociation energy of the water dissociation step but not the activation energy of the H₂ formation step. This, we believe, is the reason why the Cu-oxide interface was always proposed as the active site in all previous work.

The comments of the reviewer, whether the active sites of Cu/ZrO₂, Cu/CeO₂, Cu/TiO₂ catalysts active in the WGS reaction is Cu-oxide interface or the Cu-hydroxylated oxide interfaces, is intriguing, however, unfortunately we cannot reply because there has been not enough evidence in previous work. We will seriously consider the reviewer's comments to continue our work on other oxide/Cu catalysts in the WGS reaction in the future in order to figure out whether the nature of the active site varies with the oxide or not.

In reply to the reviewer, we have discussed this issue in the revised manuscript as the following: **"In addition to Cu/ZnO based catalysts, Cu/ZrO₂, Cu/CeO₂ and Cu/TiO₂ were also reported active in the WGS reaction, in which the Cu-oxide ensemble was proposed as the active site.³⁹⁻⁴² Our results of the Cu-hydroxylated ZnO ensemble, instead of the Cu-ZnO ensemble, as the active site of Cu/ZnO based catalysts suggest that further studies are needed to provide combined experimental and theoretical calculation evidence on the rate-limiting step of the WGS reaction, H₂O dissociation or H₂ production, in order to unambiguously identify the active sites of other Cu/oxides catalysts for the WGS reaction."** (Please see Lines 227-234). The papers recommended by the reviewer (*Int. J. Hydrogen Energ.* 39, 317-324 (2014) and *Catal. Today* 143, 45-50 (2009)) and two other highly-relevant paper (*J. Phys. Chem. C* 113, 7364-7370 (2009) and *Angew. Chem. Int. Ed.* 121, 8191-8194 (2009)) were cited as Ref. 39-42, and all references were re-ordered accordingly.

Comment 2. The authors investigate the CuZnO system for the hydrogenation of CO. However, it is not mentioned that in literature other catalytic systems such as Cu/MgO are described to be more suitable for the CO₂ free synthesis of methanol [1,4]. To study the electronic promotion of ZnO and to separate this from the support effect ZnO introduces in the classical CuZnAl system, a model system with Cu nanoparticles supported on irreducible MgO was developed by Studt *et al* [1]. The Cu/MgO catalyst showed a high methanol formation rate in a CO/H₂ mixture. This result confirms that Cu is, in general, capable of converting CO to methanol at industrially relevant rates.

Furthermore, the addition of 5 wt% ZnO to the Cu/MgO catalyst by impregnation provided evidence that the ZnO promoter plays a decisive role for the catalytic properties of Cu in industrial catalysts, as was observed for model catalysts. Comparing the different catalyst, here DFT calculations already showed that when Zn is introduced into the Cu steps all intermediates and transition states that bind through an oxygen atom are stabilized.

Author Reply: We appreciate the reviewer's insightful comments very much. The paper recommended by the reviewers used Cu/MgO model catalysts to demonstrate that Cu NPs supported on irreducible oxide was capable of catalyzing CO hydrogenation to methanol while the ZnO promoter not only greatly enhanced the catalytic activity but also changed the reaction mechanism. In our work, our large Cu nanocrystals showed high selectivity to CH₄ under our reaction conditions while ZnO/Cu NCs catalysts exhibited Cu structure-dependent catalytic selectivity in methanol formation, thus we focused on the structural effects of Cu surface on the formation and catalytic performance of CuZn alloy sites during the DFT calculations and did not consider the reaction mechanism of methanol formation on Cu surface. We will consider the reviewer's comments to carry out DFT calculation of CO hydrogenation to methanol and methane on different Cu and CuZn sites to seek for a comprehensive understanding.

In reply to the reviewer, we have discussed the commented issue in the revised manuscript as the following: **"Using Cu/MgO model catalysts,^{19, 23} it was demonstrated that Cu nanoparticles supported on irreducible oxide was capable of catalyzing CO hydrogenation to methanol while the ZnO promoter not only greatly enhanced the catalytic activity but also changed the reaction mechanism."** (Please see Line 56-60). We have also cited the suggested papers as Ref. [19,23] and reordered all the references accordingly.

Comment 3 & 4. In the manuscript the authors describe the formation of an active site for hydrogenation of CO towards methanol at the interface of an defective Cu(100) surface and Zn leading to an alloy. The description of the active site as an alloy of a defective Cu surface and Zn has already been described in literature [5,6]. It is reported to be the active site for the synthesis of methanol from hydrogenation of CO as well as CO₂ over Cu/ZnO/Al₂O₃ catalyst, however CO₂ hydrogenation is described to be significantly faster over Cu/ZnO based catalysts [1].

Since CO₂ is one of the major byproducts produced in the given work (Table S7), the formation of CO₂ and the influence of CO₂ has to be discussed in the manuscript, but it is not mentioned. Also, the origin of CO₂ is not mentioned at all.

Author Reply: We appreciate the reviewer's insightful comments very much. Previous results did show that with the similar CO and CO₂ amounts in the reactants, CO₂ hydrogenation to methanol over Cu/ZnO/Al₂O₃ proceeded faster than CO hydrogenation to methanol. Meanwhile, the methanol is believed to form via CO stepwise hydrogenation process in CO hydrogenation reaction (Ref. 17: *Science* **336**, 893-897 (2012)) and via the formate intermediate in CO₂ hydrogenation reaction (Ref. 19: *ChemCatChem* **7**, 1105-1111 (2015)). In our work, we employed the reactants of CO+H₂ (CO:H₂=1:2) at 2 MPa. Hydrocarbons were produced, accompanied by H₂O, which, reacted with CO at the Cu-hydroxylated ZnO ensemble to produce CO₂. But the amount of CO₂ was significantly less than that of CO. We calculated the highest CO₂ production amounting to only 0.0384 MPa, whereas the corresponding amount of remaining CO was 0.611 MPa; meanwhile, the in situ DRIFTS spectra observed only the CH₃O* intermediate. Thus, CO hydrogenation should be

the dominant pathway to produce methanol in our experiments.

In reply to the reviewer, we have clarified this issue in the revised manuscript as the following: **“CO₂ was always produced via the WGS reaction at the Cu-hydroxylated ZnO sites during CO hydrogenation reaction over our ZnO/Cu-NCs catalysts, in which H₂O resulted from the reactions of CO hydrogenation to hydrocarbons. CO₂ hydrogenation was reported to proceed faster than CO hydrogenation to produce methanol over Cu/ZnO/Al₂O₃ catalysts.¹⁹ However, CO hydrogenation is the dominant pathway to produce methanol over our ZnO/Cu-NCs catalysts. On one hand, the amount of produced CO₂ was significantly less than that of CO in the reaction atmosphere; on the other hand, the in situ DRIFTS spectra (Fig. 7a) only observed the CH₃O_a species but not formate species, the key intermediate respectively in CO and CO₂ hydrogenation pathways.^{17,19}”** (Please see Lines 315-323). The paper [1] suggested by the reviewer was added as Ref. [19], while the paper [5, 6] have been already cited as Ref. [13, 17] in our original manuscript. We have also re-ordered all the references accordingly.

Comment 5. Not only the conversion of CO, but also the selectivity towards methanol is significantly lower compared with the industrial methanol synthesis catalyst for all synthesized catalysts. A lower conversion is comprehensible due to the lower amount of Cu surface atoms, however is there a suitable explanation for the difference in the selectivity?

Author Reply: We appreciate the reviewer’s insightful comments very much. In our work, we used large Cu nanocrystals as the substrates to support ZnO. Comparing the industrial Cu/ZnO/Al₂O₃ catalysts, the density of surface Cu atoms are much less, leading to the decreased CO conversion; meanwhile, the density of defective Cu sites beneficial for the CuZn alloy formation are much less, leading to the decreased methanol selectivity, because the existing bare Cu sites and Cu-hydrogenated ZnO sites respectively catalyze the CO hydrogenation to hydrocarbons and WGS reactions.

In reply to the reviewer, we have discussed this issue in the revised manuscript as the following: **“Comparing the industrial Cu/ZnO/Al₂O₃ catalysts, the density of surface Cu atoms are much less, leading to the decreased CO conversion; meanwhile, the density of defective Cu sites beneficial for the CuZn alloy formation are much less, leading to the decreased methanol selectivity, because the existing bare Cu sites and Cu-hydrogenated ZnO sites respectively catalyze the CO hydrogenation to hydrocarbons and WGS reactions.”** (Please see Lines 336-342).

Comment 6. Information about the progress of CO conversion over time for the CO hydrogenation similar to the information given for the WGS (Figure S14-Figure S19) would also add additional information about how fast the restructuring happens and about the stability of the catalyst.

Author Reply: We appreciate the reviewer’s kind suggestions very much. In CO hydrogenation reaction, as-synthesized ZnO/Cu₂O catalysts were pretreated by 5%H₂/Ar at 523 K for 2h to prepare corresponding ZnO/Cu catalysts, and then the catalysts were evaluated at 523 K. The outlet gas was analyzed every 2.5 hours until the steady-state was reached.

In reply to the reviewer, we have added the catalytic performance of the representative 9%ZnO/c-Cu-34 catalyst as a function of reaction time in CO hydrogenation to methanol reaction as Supplementary Fig. 30 in the revised supporting information and discussed this issue in the revised manuscript as the following: **“The catalytic performance of representative 9%ZnO/c-Cu-34**

catalyst as a function of reaction time (Supplementary Fig. 30) suggests an in situ formation of active site, leading to increased CO conversion and CH₃OH selectivity.” (Please see Lines 244-246).

Supplementary Figure 30. The catalytic performance of 9%ZnO/c-Cu-34 catalyst as a function of reaction time in CO hydrogenation reaction at 523 K. The results display that the initial CO conversion and CH₃OH selectivity are lower (reaction time = 2.5 h) and reach the maximum at 5 h, subsequently, the catalytic performance slightly decrease. This demonstrates that the restructuring occurs during the CO hydrogenation reaction for 5 h to in situ formed CuZn alloy. The subsequent decrease of catalytic performance is likely due to the slight catalyst deactivation.

Comment 7. Furthermore, the pressure of 2 MPa applied for methanol synthesis is significantly lower compared to the pressure of 5-10 MPa, which is usually used in industry leading to a certain lack of comparability to relevant conditions.

Author Reply: We appreciate the reviewer’s insightful comments very much. The pressures of 5-10 MPa are commonly used for CO hydrogenation reaction. The aim of our work is to fundamentally understand the structural effects of Cu on Cu-ZnO interfaces for the WGS and CO hydrogenation reactions. Thus we chose a pressure of 2 MPa because the in situ characterizations could be applied at such a pressure. We agreed with the reviewer that the different pressure might lead to certain a gap between our results and the industrial conditions, but we believe that the acquired fundamental understanding can be extended to the industrial catalytic reactions.

In reply to the reviewer, we have clarified this issue in the revised supporting information as the following: “**The reaction pressure of 2 MPa was chosen in order to match with the maximum pressure of in situ characterizations.**” (Please see Lines 120 and 121 in the revised supporting information).

Comment 8. Although the authors claim plausibly that the interface between defective Cu and Zn is most important for the activity in CO hydrogenation, information about the Cu-surface area for the investigated catalysts is crucial. Metallic Cu plays a major role for activation of H₂ via dissociative adsorption. Additionally, by comparing with the BET surface area before the addition of ZnO, the Cu surface area indirectly yields information about the dispersion of the added amount of ZnO.

Author Reply: We appreciate the reviewer's kind suggestions very much. We have measured the BET surface areas of c-Cu-682, c-Cu-109, c-Cu-34, o-Cu, and d-Cu as 1.51, 2.50, 6.59, 4.09 and 3.33, respectively. We also followed the reviewer's suggestion to plot the optimized ZnO loading of ZnO/c-Cu catalysts in highly-dispersed form, which can be taken as an indicator for the as a function of the BET surface areas of c-Cu NCs (shown below, Supplementary Fig. 41). The results show that the optimized ZnO loading increases with the Cu surface area, indicating that the dispersion of supported ZnO depends on the surface area of Cu substrate.

In reply to the reviewer, we have discussed this issue in the revised manuscript as the following: **"Reasonably, the dispersion of supported ZnO in ZnO/c-Cu catalysts increases with the surface area of c-Cu substrate. (Supplementary Fig. 41)"** (Please see Lines 269-271). The BET surface areas of c-Cu-682, c-Cu-109, c-Cu-34, o-Cu, and d-Cu are added into in Supplementary Table 1 and the plot of optimized ZnO loading of ZnO/c-Cu catalysts in highly-dispersed form as a function of the BET surface areas of c-Cu NCs is added as Supplementary Fig. 41 in the revised supporting information. We have also re-ordered all Supplementary figures in the manuscript.

Supplementary Figure 41. The optimized ZnO loading of ZnO/c-Cu catalysts in highly-dispersed form as a function of the BET surface areas of c-Cu NCs. The maximum ZnO loading of ZnO/c-Cu catalysts in highly-dispersed form was derived based on the H_2 -TPR results in Supplementary Fig. 39, which is 0.99, 3.97, and 7.80 wt.% for ZnO/c-Cu-682, ZnO/c-Cu109, and ZnO/c-Cu-34, respectively.

Comment 9. The manuscript did not manage to include all relevant figures in the text in a manner it is still possible to read. Since important figures were placed in the SI part (e.g. Fig. S45), while way too much focus was laid on e.g. TEM images (Fig. 1) the manuscript is only barely readable.

Author Reply: We appreciate the reviewer's kind suggestions very much. In the revised manuscript, we have divided the original Fig. 1 into Fig. 1 (TEM images) and Fig. 2 (in situ DRIFTS spectra of CO adsorption) and included the original Fig. S45 as Fig. 7d. Meanwhile, we have also re-plotted other figures to make them easily readable.

Comment 10. XPS and IR graphs are normally plotted from high to low values.

Author Reply: We appreciate the reviewer's kind suggestions very much and have re-plotted the

relevant figures in the revised manuscript and supporting information.

Comment 11. Line 132 and 136: Figure 2C and 2D are mixed up.

Author Reply: We appreciate the reviewer's careful readings and corrected the typos in the revised manuscript (Please see Lines 149 and 153).

Comment 12. Line 214: It is not straightforward to plot two different XPS measurements within the same y axis. For a direct comparison the intensity, which depends on various factors like position of the sample inside the chamber varies to much between two consecutive measurements.

Author Reply: We appreciate the reviewer's kind suggestions very much. The different NAP-XPS measurements have been plotted separately and then merged together in the revised manuscript (Please see Fig. 4e). During a series of measurements of the same catalyst under various conditions, the sample position and the parameters of XPS spectrometer were unchanged.

Comment 13. Line 217: The authors discuss the selectivity of methanol formation, whereas the respective figure (Figure 4F) only provides information about the productivity.

Author Reply: We appreciate the reviewer's insightful comments very much. The selectivity of methanol formation is the ratio of CO-to-methanol reaction rate over the total CO reaction rate. In the original Fig. 4F and now Fig. 7c in the revised manuscript, we aim to demonstrate that $\text{Cu}_{\text{Cu}(611)}\text{Zn}$ alloy is the active site for methanol production, thus it is reasonable that the productivity of methanol, instead of the methanol selectivity, should be used to correlate to the number of surface sites on $\text{Cu}_{\text{Cu}(611)}\text{Zn}$ alloy in ZnO/c-Cu catalysts, represented by the intensity of CO absorbed on $\text{Cu}_{\text{Cu}(611)}\text{Zn}$ alloy.

Comment 14. Line 238: Please discuss that there is a significant pressure gap between the catalytic measurements and the DRIFTS and in situ transmission FT-IR spectra. Additionally, even if the pressure is mentioned in Fig. 4D, in the figures and table in the supportive information a pressure is not noted.

Author Reply: We appreciate the reviewer's careful readings very much. We are very sorry for the confusion commented by the reviewer. As described in the manuscript (Please see Lines 283-314) and in the supporting information (Please see Lines 66-95), two sets of in-situ infrared measurements, in situ DRIFTS and in situ transmission FT-IR, were performed for CO hydrogenation reactions. The in situ DRIFTS spectra during CO hydrogenation reaction shown in Fig. 7a, Supplementary Figs. 42-45 and Supplementary Table 8 were used to probe surface intermediates, whereas the in situ transmission FT-IR spectra with an in situ reactor for transmission FT-IR measurements under conditions varying from high pressures and high temperatures to vacuum and low temperatures was used to firstly characterize various catalysts under CO hydrogenation reaction and then probe their structures by CO adsorption at low temperature without exposures of the used catalysts to air. The acquired in situ transmission FT-IR spectra under CO hydrogenation reaction are similar to the corresponding in situ DRIFTS spectra (Supplementary Fig. 46). Then the reactor was evacuated and collected down to 123 K, and CO adsorption with $P_{\text{CO}}=1000$ Pa was measured in order to probe surface structures of used catalysts (Fig. 7b and Supplementary Fig. 47). Although the conditions of catalyst under CO adsorption and under CO hydrogenation reaction are different, using our strategy, in situ formed CuZn alloys during CO hydrogenation can be

prevented from surface oxidizing and reliably characterized. Our results also demonstrated in situ formed CuZn alloys as the active site for CO hydrogenation to methanol. As far as we know, our in situ IR results represent the first spectroscopic evidence to identify in situ formed CuZn alloys as the active site for CO hydrogenation to methanol catalyzed by Cu/ZnO catalysts.

In reply to the reviewer, we have re-written the relevant texts to clarify the issue in the revised manuscript as the following: **“An in situ reactor for transmission FT-IR measurements under conditions varying from high pressures and high temperatures to vacuum and low temperatures was used to firstly characterize various catalysts under CO hydrogenation reaction and then probe their structures by CO adsorption at low temperature without exposures of the used catalysts to air.”** (Please see Lines 290-294); we have also included the measurement conditions in the figure captions and table titles in the revised manuscript and supporting information.

Comment 15. Line 251: Describing Fig. 4E and Fig. S43 please add a naming of the different fits in blue, red and light blue.

Author Reply: We appreciate the reviewer’s kind suggestions very much. We have added a naming of the different fitted line in the captions of Fig. 7, Supplementary Figure 47 and other figures in the revised manuscript and supporting information.

Comment 16. Line 261: Since the reliability the quantification of IR spectra strongly depends on the chosen region due to possible overlapping bands, here a table of the chosen limits while integrating the bands for Fig. 4F and S45 would be straightforward.

Author Reply: We appreciate the reviewer’s kind suggestions very much. During the peak fitting processes of IR spectra, we used the minimum fitted peaks with the same peak shape and similar FWHM. Then the integrated peak areas of the fitted peaks were used to correlate with densities of corresponding surface sites.

In reply to the reviewer, we have listed all parameters and results of peak-fitting processes in supplementary Table 2 and 10 in the revised supporting information, and mentioned them in the revised manuscript (Please see Lines 101, 102, 114, 115 and 298). We have also re-ordered other supplementary Tables in the supporting information.

Comment 17. Line 424: This paper is from 2020 not 2011.

Author Reply: We appreciate the reviewer’s careful readings very much and have corrected the typo in the revised manuscript.

Comment 18. Figure 4A is misleading, because the selectivities of CO₂ are not included.

Author Reply: We appreciate the reviewer’s kind suggestions very much. We have clarified this issue by adding **“CO₂ is not included for selectivity calculations”** in the caption of Fig. 6 (the original Fig. 4) in the revised manuscript.

Comment 19. SI Line 370: Fig. S14: “most stable CO species adsorbed on...”

Author Reply: We appreciate the reviewer’s careful readings and have corrected to “stable CO species adsorbed on” in the caption of Fig. S14 in the revised supporting information.

References suggested by the reviewer

- [1] F. Studt, M. Behrens, E. L. Kunkes, N. Thomas, S. Zander, A. Tarasov, J. Schumann, E. Frei, J. B. Varley, F. Abild-Pedersen et al., *ChemCatChem* 2015, 7, 1105–1111.
- [2] C. Chen, C. Ruan, Y. Zhan, X. Lin, Q. Zheng, K. Wei, *Int. J. Hydrogen Energy* 2014, 39, 317–324.
- [3] J. A. Rodriguez, P. Liu, X. Wang, W. Wen, J. Hanson, J. Hrbek, M. Pérez, J. Evans, International Symposium on Catalysis for Ultra-Clean Fuels, Dalian, China, July 21-24, 2008 2009, 143, 45–50.
- [4] N. D. Nielsen, J. Thrane, A. D. Jensen, J. M. Christensen, *Catal Lett* 2020, 150, 1427–1433.
- [5] F. Studt, M. Behrens, F. Abild-Pedersen, *Catal. Lett. (Catalysis Letters)* 2014, 144, 1973–1977.
- [6] M. Behrens, F. Studt, I. Kasatkin, S. Kühl, M. Hävecker, F. Abild-Pedersen, S. Zander, F. Girgsdies, P. Kurr, B.-L. Kniep et al., *Science* 2012, 336, 893–897.

Authors' Reply to Reviewer 2's Comments and Revisions

Comment: The current study reports the mechanistic understanding of active sites for WGS and CO hydrogenation using the combined experiment-theory studies. Major revisions are suggested to be made to improve the quality and merit the publication in *Nature Commun.*

Author Reply: We appreciate the reviewer's positive recommendation and valuable comments very much. We have seriously considered the reviewer's comments and revised the manuscript accordingly. We hope that the revised manuscript will be suitable for the publication in *Nature Communications*.

Comment 1: First of all, the catalytic process selected here is different from that has been extensively studied previously for Cu-ZnO: CO₂ hydrogenation to methanol. This is the author-called "long-existing debate" with lots of cited papers. The change from CO₂ to CO hydrogenation imposes different effects on reaction mechanism and surface structures, which are the focus here. Thus, to target the "long-existing debate", as claimed, CO₂ rather than CO hydrogenation should be studied. Otherwise, the provided understanding may not be meaningful.

Author Reply: We appreciate the reviewer's insightful comments very much. We agree with the reviewer that CO₂ hydrogenation catalyzed by Cu/ZnO catalysts is now an important and hot topic due to the serious "greenhouse effect" caused by the emission of CO₂. However, Cu-ZnO-Al₂O₃ catalysts have been traditionally used as the industrial catalysts for both WGS and CO hydrogenation to methanol reactions. The relevant fundamental studies have been extensively carried out, but debates still exist on the nature of active sites due to the lack of solid experimental evidence and the active structures of Cu in both types of Cu-ZnO-Al₂O₃ catalysts are not established. Thus, we initiated our approach of using ZnO/Cu-NCs catalysts to comprehensively study the structure-activity relations in both WGS and CO hydrogenation to methanol reactions in order to identify the active sites and reaction mechanisms six years ago. To be honest, we are now using the similar approach to study the structure-activity relation of Cu-NCs and ZnO/Cu-NCs in CO₂ hydrogenation to methanol. The preliminary results are interesting, and we hope that we will be able to present the results in the future.

In reply to the reviewer, we have added another review on Cu/ZnO/Al₂O₃ catalysts for CO hydrogenation to methanol (*Adv. Catal.* **31**, 243-313, (1982)) as Ref. 9 in the revised manuscript to support that Cu/ZnO/Al₂O₃ catalyzed CO hydrogenation to methanol reaction is an important industrial reaction with a long history (Please see Line 43 and Ref. 9).

Comment 2: For CO adsorption on Cu, no CO-Cu(1+) interaction was observed by IR. Is it possible that CO results in the reduction of Cu(1+) to Cu(0), which may occur on the surface?

Author Reply: We appreciate the reviewer's insightful comments very much. Our CO adsorption experiments were carried out at 123 K, whose DRIFTS spectra were in situ recorded. At such a low temperature, reduction of Cu(I) by CO unlikely occurs.

In reply to the reviewer, we have indicated the CO adsorption conditions in the captions of relevant figures in the revised manuscript and supporting information.

Comment 3: In DFT calculations, a nanorow of ZnO was deposited on Cu(111). Was there any strain

artificially introduced on ZnO layer to accommodate the lattice mismatch between ZnO and Cu? How will that affect the results? Is the nanorow good enough to describe the experimental samples?

Author Reply: We appreciate the reviewer's insightful comments very much. As described in the Supporting Information (Please see Lines 149-152), the models of ZnO/Cu, one-layer graphite-like (1×4) ZnO(0001) ribbon, with an in-plane lattice of 3.30 Å, on three-layer (5×6) Cu(111) and three-layer (5×6) Cu(100) slab, were adopted to simulate Cu/ZnO interface according to previous results [8-12] (*Angew. Chem., Int. Ed.* **54**, 4544-4548 (2015); *J. Phys. Chem. C* **117**, 11211-11218 (2013); *Angew. Chem., Int. Ed.* **52**, 11925-11929 (2013); *Phys. Rev. Lett.* **99**, 026102-026105 (2007); *J. Phys. Chem. C* **114**, 15432-15439 (2010)). The interface of ZnO and Cu is simulated by 5 Cu units and 4 ZnO units along the ZnO periodic direction and the mismatch between 5Cu (12.7799Å) and 4ZnO (12.9971Å) is only 1.5%, which is within the calculation error of DFT. We also made a simple test by extending the model from (1×4) ZnO(0001) on (5×6) Cu used in our study to (2×4) ZnO(0001) on (5×8) Cu and did not find much difference.

In reply to the reviewer, we have clarified this issue in the revised Supporting Information as the following: **"The lattice mismatch at the ZnO-Cu interface of 5Cu (12.7799 Å) and 4ZnO (12.9971 Å) is 1.5%, within the calculation error of DFT."** (Please see Lines 152 and 153).

Comment 4: How was *OH coverage defined? in DFT calculations, it seems to based on number of Zn site, which is 0.75ML. I am not sure how 0.75ML of *OH on ZnO was determined. Is it consistent with experiment? Will the coverage of *OH change the catalytic behaviors?

Author Reply: We appreciate the reviewer's insightful comments very much. *OH coverage was defined by the ratio of OH number against the total O number at the Cu-ZnO interface in the used models. As shown in Fig. 5, we increased the OH number at the Cu-ZnO interface from 0 to 3, giving OH* coverages of 0, 0.25, 0.5, and 0.75 ML because the total number of O at the Cu-ZnO interface in the used models is 4. DFT calculation results show that the H₂ production step is the rate-limiting step for the surfaces with OH* coverage up to 0.5 eV while the H₂O dissociation becomes the rate-limiting step for the surface with 0.75 ML OH*. Experimentally, we used CO+H₂O TPRS results to demonstrate that H₂ production is not unlikely to be the rate-limiting step of WGS reaction catalyzed by our ZnO/Cu-NCs catalysts. Thus we concluded that the Cu-hydroxylated ZnO interface, instead of the Cu-ZnO interface, should be the active site of Cu/ZnO catalysts to catalyze the WGS reaction. However, we could not experimentally determine the accurate OH* coverage at the Cu-ZnO interface of the Cu/ZnO catalysts during the WGS reaction because the OH and O signals in the XPS spectra were from the entire surface, instead of the Cu-ZnO interface. Based on the present DFT calculation results, we can surely conclude that the critical OH* coverage at which the rate-limiting step changes from the H₂ production to the water dissociation is above 0.5 ML. By adopting large models in the DFT calculations, the critical OH* coverage at the Cu-ZnO interface at which the rate-limiting step changes from the H₂ production to the water dissociation can be further refined, but this needs a substantial DFT calculation work and is not the task of the present work. Therefore, we only used the accurate OH* coverage when describing the DFT calculation results, but only used hydroxylated-ZnO elsewhere in the manuscript.

In reply to the reviewer, we have re-written the texts to clarify this issue in the revised manuscript as the following: **"We thus calculated the activation energy of water dissociation and H transfer reaction at the OH-covered Cu-ZnO interfaces, in which the OH coverage is defined as the ratio**

of OH number against total O number at the ZnO-Cu interface. The calculated activation energy of water dissociation was found to increase with the OH coverage at the Cu-ZnO interface while the calculated activation energy of H transfer reaction to decrease (Fig. 5c, Supplementary Fig. 28 and Supplementary Table 6), but water dissociation still exhibits smaller activation energy than H transfer reaction at the ZnO-Cu interfaces with OH coverages up to 0.5 ML. When the OH coverage increases to 0.75 ML, the activation energy of H₂O dissociation increases to 1.05 and 0.87 eV respectively at the 0.75 ML OH_{ZnO}-ZnO-Cu(111) (Fig. 5b1) and 0.75 ML OH_{ZnO}-ZnO-Cu(100) (Fig. 5b2) interfaces, larger than the corresponding activation energy of subsequent H transfer reaction, being 0.88 and 0.76 eV, respectively. These DFT calculation results suggest that the rate-limiting step of WGS reaction changes from the H transfer reaction, i.e. the H₂ production, at the ZnO-Cu interfaces with OH coverages up to 0.5 ML to water dissociation at the ZnO-Cu interface with an OH coverage of 0.75 ML.” (Please see Lines 205-220). The definition of OH coverage is also indicated in the caption of Fig. 5 (Please see Lines 598 and 599).

Authors' Reply to Reviewer 3's Comments and Revisions

Comment: The present paper by Zhang et al. is an interesting study of the active sites of Cu/ZnO catalysts in the WGS reaction and in the methanol formation by CO hydrogenation. The authors combine an elegant synthetic approach, namely the preparation of Cu₂O pre-catalyst with defined shape and termination that were after the addition of ZnO transformed into Cu metal while maintaining the particle morphology of the inverse catalyst type. Many complementary techniques have been used to explain the catalytic data. The major conclusions of the regarding the active sites are based on CO-DRIFTS and DTF calculation. The results are certainly of high interest to the catalysis community and deserve publication.

Author Reply: We appreciate the reviewer's positive and valuable comments very much. We have seriously considered the reviewer's comments and revised the manuscript accordingly. We hope that the revised manuscript will be suitable for the publication in *Nature Communications*.

Comment 1: However, I do not see that these results "conclude the long-existing debate" on the active sites of this catalyst as claimed by the authors in the abstract. Both the special Cu-ZnO perimeter site and the stepped CuZn alloy site were suggested previously, and this paper presents support for their role in WGS and methanol synthesis, respectively. The suggested Cu-hydroxylated ZnO at Cu(100) ensembles and the CuZn at Cu(661) alloy may be new specific suggestions in the debate, but the experimental evidence presented for their existence compared to other possible sites on the non-uniform powders is rather limited.

Therefore, I recommend revising the final claim of unambiguous identification of these sites and submission to a journal specialized on chemistry or catalysis.

Author Reply: We appreciate the reviewer's insightful comments very much. As discussed in the introduction part, identifying the active sites of Cu-ZnO-Al₂O₃ catalysts industrially used in the WGS and CO hydrogenation reactions are of great interest and importance. The Cu-ZnO interface and CuZn alloys at the stepped Cu sites were proposed as the active sites for the WGS and CO hydrogenation reactions, respectively, but debates still exist. In our work, via combined experimental and theoretical studies of a series of ZnO/Cu-NCs catalysts, we provide solid evidence to demonstrate the Cu₁₀₀-hydroxylated ZnO ensemble and Cu₍₆₁₁₎Zn alloys as the active sites for the WGS and CO hydrogenation reactions, respectively. We agree with the reviewer that the results of our ZnO/Cu-NCs might not be simply extended to the commercial non-uniform Cu/ZnO/Al₂O₃ catalysts, but they do provide novel insights in the active sites of Cu/ZnO-based catalysts for the WGS and CO hydrogenation reactions and reveal the Cu structural effect. Our results also demonstrate that the active site of a catalyst varies with the reaction being catalyzed, which, we believe, is an important concept. Meanwhile, our work spans heterogeneous catalysis, energy chemistry, surface chemistry and theoretical chemistry. Therefore, it is suitable for the publication in *Nature Communication*.

In reply to the reviewer, we have re-written the sentence commented by the reviewer in the revised manuscript as the following: "These results provide novel insights in the active sites of Cu-ZnO catalysts for the WGS and CO hydrogenation reactions and reveal the Cu structural effects, and offer..." (Please see Lines 33-35).

Before re-submission, I recommend to consider the following aspects when revising the paper:

Comment 2: Some of the reactivity data deviates from expectations for Cu-based catalysts. For example, the dominant formation of methane or the high selectivity to C_2^+ products in the methanol synthesis on Cu/ZnO are surprising. This could lead to the conclusion that the catalysts prepared are substantially different from industrial catalysts and may not be suitable models for the elucidation of the active sites of Cu/ZnO/Al₂O₃, which shows almost 100% selectivity to methanol.

Author Reply: We appreciate the reviewer's insightful comments very much. In our work, we used large Cu nanocrystals as the substrates to support ZnO. Comparing the industrial Cu/ZnO/Al₂O₃ catalysts, the density of surface Cu atoms are much less, leading to the decreased CO conversion; meanwhile, the density of defective Cu sites beneficial for the CuZn alloy formation are much less, leading to the decreased methanol selectivity, because the existing bare Cu sites and Cu-hydrogenated ZnO sites respectively catalyze the CO hydrogenation to hydrocarbons and WGS reactions. However, we believe that the acquired fundamental understanding of the active site and Cu structural effects of Cu/ZnO catalysts in CO hydrogenation to methanol can be reasonably extended to the industrial Cu/ZnO/Al₂O₃ catalysts.

In reply to the reviewer, we have discussed this issue in the revised manuscript as the following: **"Comparing the industrial Cu/ZnO/Al₂O₃ catalysts, the density of surface Cu atoms are much less, leading to the decreased CO conversion; meanwhile, the density of defective Cu sites beneficial for the CuZn alloy formation are much less, leading to the decreased methanol selectivity, because the existing bare Cu sites and Cu-hydrogenated ZnO sites respectively catalyze the CO hydrogenation to hydrocarbons and WGS reactions."** (Please see Lines 336-342).

Comment 3: Why were the edges or corners of the particles not considered as possible active sites? Given the established structure-sensitivity of both reactions, these seem likely candidates, which should not be neglected in the discussion of the active sites and in the assignment of CO chemisorption data.

Author Reply: We appreciate the reviewer's insightful comments very much. In our work, we used CO adsorption, which is very sensitive to the coordination environments of Cu, to probe the surface structures of Cu NCs and ZnO/Cu-NCs. As described and discussed in Lines 96-127, c-Cu NCs show two vibrational bands at 2085 and 2101-2106 cm⁻¹ arising from CO adsorbed respectively at the terrace and defective sites of Cu{100} facets, o-Cu NCs show two vibrational bands at 2075 and 2107 cm⁻¹ arising from CO adsorbed respectively at the terrace and defective sites of Cu{111} facets, and d-Cu NCs show one vibrational band at 2093 cm⁻¹ arising from CO adsorbed at the terrace sites of Cu{110} facets. We failed to observe CO adsorbed at the edge sites, for examples, of c-Cu which should exhibit the {110} orientation. These results suggest that compared with the density of face sites, the density of edge and corner sites is very low. Therefore, we did not consider their contributions to the catalytic performance. This is supported by experimental observations that all ZnO/c-Cu catalysts exhibit similar apparent activation energy in the WGS reaction, suggesting the dominant contribution from the Cu{110} face sites.

In reply to the reviewer, we have discussed this issue in the revised manuscript as the following: **"Vibrational features of CO adsorbed at the edge or corner sites were hardly observed,**

particularly for the c-Cu-34 and ZnO/c-Cu-34 catalysts, indicating that their density should be much lower than the density of face sites” (Please see Lines 125-127) and “also indicating that the face sites of Cu NCs in ZnO/c-Cu catalysts dominantly contribute to the catalytic activity,” (Please see Lines 138 and 139).

Comment 4: Activity data for the methanol synthesis should be provided. CO hydrogenation on CuZn is typically much slower than CO₂ hydrogenation that could happen in CO/CO₂/H₂ syngas, which should be discussed.

Author Reply: We appreciate the reviewer’s insightful comments very much. We have provided the activity data in Supplementary Table 8 in our original manuscript, in addition to in Fig. 6 a1-a4 (Please see Lines 235 and 236). Previous results did show that with the similar CO and CO₂ amounts in the reactants, CO₂ hydrogenation to methanol over Cu/ZnO/Al₂O₃ proceeded faster than CO hydrogenation to methanol. Meanwhile, the methanol is believed to form via CO stepwise hydrogenation process in CO hydrogenation reaction (Ref. 17: *Science* **336**, 893-897 (2012)) and via the formate intermediate in CO₂ hydrogenation reaction (Ref. 19: *ChemCatChem* **7**, 1105-1111 (2015)). In our work, we employed the reactants of CO+H₂ (CO:H₂=1:2) at 2 MPa. Hydrocarbons were produced, accompanied by H₂O, which, reacted with CO at the Cu-hydroxylated ZnO ensemble to produce CO₂. But the amount of CO₂ was significantly less than that of CO. We calculated the highest CO₂ production amounting to only 0.0384 MPa, whereas the corresponding amount of remaining CO was 0.611 MPa; meanwhile, the in situ DRIFTS spectra observed only the CH₃O* intermediate. Thus, CO hydrogenation should be the dominant pathway to produce methanol in our experiments.

In reply to the reviewer, we have clarified this issue in the revised manuscript as the following: “CO₂ was always produced via the WGS reaction at the Cu-hydroxylated ZnO sites during CO hydrogenation reaction over our ZnO/Cu-NCs catalysts, in which H₂O resulted from the reactions of CO hydrogenation to hydrocarbons. CO₂ hydrogenation was reported to proceed faster than CO hydrogenation to produce methanol over Cu/ZnO/Al₂O₃ catalysts.¹⁹ However, CO hydrogenation is the dominant pathway to produce methanol over our ZnO/Cu-NCs catalysts. On one hand, the amount of produced CO₂ was significantly less than that of CO in the reaction atmosphere; on the other hand, the in situ DRIFTS spectra (Fig. 7a) only observed the CH₃O_a species but not formate species, the key intermediate respectively in CO and CO₂ hydrogenation pathways.^{17,19}” (Please see Lines 315-323). Ref 19 is added as to support our arguments. We have also re-ordered all the references accordingly.

Comment 5: Please provide experimental evidence that not only the particle morphology but also the single crystalline nature of the particle was maintained when reducing Cu₂O, e.g. by electron diffraction like shown in Fig. S1 for the Cu₂O. For example, in Figure 1 A4, B4, the 200 lattice planes seem not oriented in parallel or perpendicular to the cube’s 100 surface.

Author Reply: We appreciate the reviewer’s kind suggestions very much. We have acquired corresponding ED patterns of TEM images of ZnO/Cu samples shown in Fig. 1 and added them into the revised Fig. 1. The results demonstrate the single crystalline nature of the Cu particles.

In reply to the reviewer, we have discussed this issue in the revised manuscript as the following: “and electron diffraction patterns indicate that all Cu NCs are single crystals,” (Please see Lines

93 and 94) and “**Insets show corresponding electron diffraction patterns of TEM images.**” (Please see Fig. 1 and Lines 566 and 567).

Comment 6: Even if 100 particles were evaluated in HRTEM, the alloy formation and its extent should be confirmed and quantified by XRD, which should be well suited for this purpose.

Author Reply: We appreciate the reviewer’s kind suggestions very much. For each sample, we counted more than 100 particles, and the highest fraction of CuZn alloy particles was around 47.2% in the 9%ZnO/c-Cu-34 sample with the highest CH₃OH selectivity. The formed CuZn alloy are fine and exhibit a size distribution of 3.8±1.0 nm, beyond the detection limit of XRD. Other samples with small fractions of fine CuZn alloy particles reasonable could not exhibit their XRD patterns. In the XRD pattern of all used catalysts exposed to air, we only observed diffraction peaks arising from Cu₂O and CuO, demonstrating the facile oxidation of Cu nanoparticles.

In reply to the reviewer, we have discussed this issue in the revised manuscript as the following: **“CuZn alloy with a size distribution of 3.8±1.0 nm was identified on used ZnO/o-Cu and ZnO/c-Cu catalysts in HRTEM images (Fig. 6 b2 and b3, Supplementary Figs. 33-37). In the corresponding XRD patterns (Supplementary Fig. 38), diffraction peaks of Cu₂O and CuO were observed, demonstrating facile oxidation of Cu nanoparticles upon exposures to air, whereas no peaks from CuZn alloy could be identified probably due to their fine size.”** (Please see Lines 252-257). Size distribution of ZnO particles in fresh and CuZn alloy in used 9%ZnO/c-Cu-34 is added as Supplementary Figure 37, and XRD patterns of used c-Cu-34 and ZnO/c-Cu-34 is added as Supplementary Figure 38. We have also re-ordered the Supplementary Figures.

Comment 7: The Miller index of the first reflection in Fig. S2 is wrong.

Author Reply: We appreciate the reviewer’s careful readings very much. We have corrected the typos in the revised supporting information (Please see Supplementary Figure 2 in the revised supporting information).

REVIEWERS' COMMENTS

Reviewer #2 (Remarks to the Author):

The responses to the comments are satisfactory and the revised manuscript is recommended for acceptance.

Authors' Reply to Reviewer 2's Comments and Revisions

Reviewer #2 (Remarks to the Author):

Comment: The responses to the comments are satisfactory and the revised manuscript is recommended for acceptance.

Author Reply: We appreciate the reviewer's recommendation very much.